# Average Sensitivity of Hierarchical Clustering

## Abstract

Hierarchical clustering is one of the most popular methods used to extract cluster structures in a dataset. However, if the hierarchical clustering algorithm is sensitive to a small perturbation to the dataset, then the credibility and replicability of the output hierarchical clustering are compromised. To address this issue, we consider the average sensitivity of hierarchical clustering algorithms, which measures the change in the output hierarchical clustering upon deletion of a random data point from the dataset. Then, we propose a divisive hierarchical clustering algorithm with which we can tune the average sensitivity. Experimental results on benchmark and real-world datasets confirm that the proposed method is stable against the deletion of a few data points, while existing algorithms are not.

## 1  Introduction

Hierarchical clustering is one of the most popular methods used to extract cluster structures in a dataset consisting of data points (Murtagh and Contreras, 2012b). This method partitions the data points into clusters by constructing a rooted tree whose leaves correspond to data points and internal nodes represent clusters. By tracing the hierarchy from the root to leaves, we can extract interpretable knowledge from the dataset. For example, suppose that we have genomic data of single cells in a tissue. Then, the hierarchy can be used to figure out complex cellular states and tissue compositions (Žurauskienė and Yau, 2016). Hierarchical clusterings are also used in several applications such as phylogenetics (Eisen et al., 1998), geophysics (Takahashi et al., 2019), and social network analysis (Gilbert et al., 2011).

Because of the importance of hierarchical clustering, a plethora of hierarchical clustering algorithms have been proposed (Heller and Ghahramani, 2005; Jain, 2010; Hastie et al., 2009; Murtagh and Contreras, 2012b). These algorithms are mainly concerned with the quality of the output hierarchical clustering. However, there is another essential aspect that must not be overlooked: stability of the output hierarchical clustering. Since the output is often used to understand the data structure, an algorithm needs to be stable to data perturbations as long as the data distribution remains intact. This requirement can be naturally formalized as a question using the notion of average sensitivity (Varma and Yoshida, 2021); given a random deletion of data points from the original dataset, how stable is the output hierarchical clustering? In the example of genomic data, a stable and reliable algorithm is expected to retain most of the tissue compositions found in the original, even if a few cells are missing. However, in the example in Figure 1 and in the application to geophysics (Figure 3 in Section 8), we show that the existing algorithms are unstable for data point removals.

In this work, we propose a novel algorithm for hierarchical clustering that is stable against deletions of data points. We measure the stability of an algorithm using average sensitivity (Murai and Yoshida, 2019; Varma and Yoshida, 2021). Because the average sensitivity was originally defined for algorithms that output vectors or sets, we first formally define the average sensitivity of hierarchical clustering algorithms. Then, we propose a (randomized) algorithm that partitions the dataset in a top-down manner. The proposed algorithm applies a randomized process called the exponential mechanism (McSherry and Talwar, 2007) when partitioning the dataset, and we theoretically prove that it has a small average sensitivity.

Figure 1 shows an illustrative example of sensitive/stable hierarchical clustering algorithms. In this example, the standard agglomerative method induces different hierarchies before and after one data

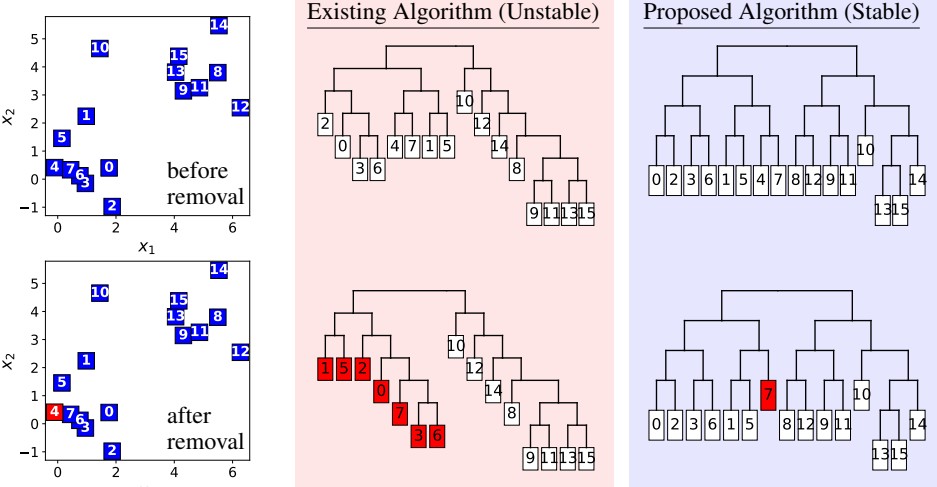

Figure 1: Examples of a dataset (top left) and its hierarchical clusterings output by an existing agglomerative algorithm using complete linkage (top middle) and the proposed one (top right), and a dataset obtained by removing the data point 4 (bottom left) and its hierarchical clusterings output by the existing agglomerative algorithm (bottom middle) and the proposed one (bottom right). The existing agglomerative clustering algorithm is sensitive to the removal of even a single data point. The proposed algorithm produces a more stable clustering. The red nodes in the right trees denote the change from the trees in the left before the data removal.

point (the data point 4) is removed, as shown in the middle of the figure. This result indicates that the widely used agglomerative method is sensitive to the removal of data points. The objective of this study is to design a hierarchical clustering algorithm that is stable against the removal of a few data points, as shown in the bottom of the figure.

Randomized algorithms may output completely different hierarchical clusterings on the original dataset and on that obtained by deleting a random data point even if the output distributions are close. To alleviate this issue, we design a (randomized) hierarchical clustering algorithm with low average sensitivity under shared randomness, which outputs similar hierarchical clusterings both on the original dataset and on the dataset obtained by deleting a random data point with a high probability over the choice of the random bits used.

We conduct comparisons between our proposed algorithm and existing algorithms with three benchmark datasets. In the experiments, we evaluated the trade-offs between the average sensitivity of the clustering algorithms and their clustering qualities. We observed that most of the existing algorithms exhibit high average sensitivity indicating that their output can change drastically even for the removal of a single data point. By contrast, the proposed algorithm can produce stable clustering results, while maintaining the quality of clustering. We also applied the clustering algorithms to a real-world GPS dataset (Takahashi et al., 2019). The results on this dataset also confirms that the existing algorithms are sensitive to data deletion, while the proposed algorithm is not.

## 2 RELATED WORK

**Hierarchical Clustering**  Algorithms for hierarchical clustering can be classified into agglomerative and divisive methods (Hastie et al., 2009). Given a dataset, an agglomerative method iteratively finds a pair of data points or clusters using a certain linkage criterion and merges them into a new cluster until all the data points are merged into a single cluster. As the linkage criterion, the single linkage, average linkage, and complete linkage rules are frequently used (Hastie et al., 2009; Murtagh and Contreras, 2012a). A divisive method constructs a hierarchy in a top-down manner. It recursively partitions a dataset into two sub-clusters until all the data points are partitioned or it reaches a prescribed tree depth (Jain, 2010).

Several extensions of the clustering algorithms are considered; Abboud et al. (2019); Moseley et al. (2021) considered improving the computational scalability; Ackerman et al. (2012) introduced a

weighted version of the agglomerative methods; and Kimes et al. (2017) and Gao et al. (2022) introduced statistical tests for clustering. Theoretical aspects of hierarchical clustering are also investigated; (Dasgupta, 2016) introduced a cost function for hierarchical clustering; Ackerman and Ben-David (2016) showed that the agglomerative methods have some desirable properties; and Roy and Pokutta (2016); Charikar and Chatziafratis (2017); Moseley and Wang (2017); Dhulipala et al. (2022) proposed methods with better approximation guarantees.

We note that the focus of the studies above is on constructing hierarchies with better quality or more efficiency. The current study is orthogonal to them; our focus is on developing a hierarchical clustering algorithm that are stable against the deletion of a data point.

**Robust Hierarchical Clustering** There have been a few studies on hierarchical clustering algorithms that exhibit robustness against outlier injections (Eriksson et al., 2011; Balcan et al., 2014; Cheng et al., 2019), which is a distinct form of data perturbation compared to the current study. These studies aim to achieve consistent clustering results regardless of the presence of outliers by identifying the injected outliers. It is important to note that hierarchical clustering algorithms can be unstable even in the absence of outliers. As demonstrated in Figure 1, although the underlying data distribution does not change after deleting a data point, the clustering results can differ significantly. For reliable knowledge discovery, it is imperative that the algorithm remains stable for such natural perturbations in the data. However, this specific type of robustness has not yet been thoroughly explored, making our study the first to venture in this direction.

**Average Sensitivity** The notion of average sensitivity was originally introduced in (Murai and Yoshida, 2019) to compare network centralities in terms of their stability against graph perturbations. Then the notion was extended to handle graph algorithms in (Varma and Yoshida, 2021). Since then average sensitivity of algorithms for various problems have been studied, including the maximum matching problem (Yoshida and Zhou, 2021), spectral clustering (Peng and Yoshida, 2020), Euclidean $k$-clustering (Yoshida and Ito, 2022), dynamic programming problems (Kumabe and Yoshida, 2022a;b), and decision tree learning (Hara and Yoshida, 2023).

## 3 PRELIMINARIES

We use bold symbols to denote random variables. For two random variables $\boldsymbol{X}$ and $\boldsymbol{Y}$ on a finite set $E$, let $d_{\mathrm{TV}}(\boldsymbol{X}, \boldsymbol{Y}) := \sum_{e \in E} |\Pr[\boldsymbol{X} = e] - \Pr[\boldsymbol{Y} = e]|/2$ denote the total variation distance between their distributions. For sets $S$ and $T$, let $S \triangle T = (S \setminus T) \cup (T \setminus S)$ denote their symmetric difference.

### 3.1 HIERARCHICAL CLUSTERING

Let $X = \{x_1, \ldots, x_n\}$ be a dataset. We always assume that the data points $x_1, \ldots, x_n$ are distinct (otherwise we assign them unique IDs so that they are distinct). A *hierarchical clustering* over $X$ is a rooted tree $T$ such that each leaf is corresponding to a subset of $X$ and the subsets corresponding to leaves form a partition of $X$. Note that hierarchical clustering considered in this work does not always decompose $X$ into data points. Let $\mathsf{root}(T)$ denote the root node of $T$. In this work, we mostly consider binary trees and let $\mathsf{left}(T)$ and $\mathsf{right}(T)$ denote the left and right, respectively, subtrees of $\mathsf{root}(T)$. If $\mathsf{root}(T)$ is the only node in $T$, then we call $T$ a *singleton*, and we define $\mathsf{left}(T) = \mathsf{right}(T) = \emptyset$. Also we set $\mathsf{left}(T) = \mathsf{right}(T) = \emptyset$ when $T$ is an empty tree. Let $\mathsf{leaves}(T) \subseteq 2^X$ denote the leaves of $T$.

### 3.2 GRAPH-THEORETIC NOTIONS

For a finite set $V$, we denote by $\binom{V}{2}$ the set of pairs of elements in $V$. For a set $V$ and $i \in V$, we sometimes write $V - i$ to denote $V \setminus \{i\}$. Let $G = (V, E)$ be a graph. For a vertex $i \in V$, let $G - i$ denote the graph obtained from $G$ by deleting $i$ and the edges incident to $i$. For a vertex set $S \subseteq V$, let $G[S]$ denote the subgraph of $G$ induced by $S$.

Let $G = (V, E, w)$ be a weighted graph, where $w : E \to \mathbb{R}_+$ is a weight function over edges. For disjoint sets of vertices $S, T \subseteq V$, let $c_G(S, T)$ denote the total weight of edges between $S$ and $T$, that is, $\sum_{i \in S, j \in T} w(i, j)$. We denote by $\phi_G(S)$ the *sparsity* of $S$, that is, $c_G(S, V \setminus S)/(|S| \cdot |V \setminus S|)$.

### 3.3 EXPONENTIAL MECHANISM

The *exponential mechanism* (McSherry and Talwar, 2007) is an algorithm that, given a vector $x \in \mathbb{R}^n$ and a real number $\lambda > 0$, returns an index $i \in [n]$ with probability proportional to $e^{-\lambda x_i}$. The following fact is useful to design algorithms with low average sensitivity.

**Lemma 3.1** (McSherry and Talwar (2007)). *Let $\lambda > 0$ and let $A$ be the algorithm that, given a vector $x \in \mathbb{R}^n$, applies the exponential mechanism to $x$ and $\lambda$. Then for any $t > 0$, we have*

$$\Pr_{i \sim A(x)} \left[ x_i \geq \mathsf{OPT} + \frac{\log n}{\lambda} + \frac{t}{\lambda} \right] \leq e^{-t},$$

*where* $\mathsf{OPT} = \min_{i \in [n]} x_i$. *Moreover, for $x' \in \mathbb{R}^n$, we have*

$$d_{\mathrm{TV}}(A(x), A(x')) = O\left( \lambda \cdot \|x - x'\|_1 \right).$$

## 4 AVERAGE SENSITIVITY OF HIERARCHICAL CLUSTERING

In this section, we formally define the average sensitivity of a hierarchical clustering algorithm.

### 4.1 DISTANCE BETWEEN HIERARCHICAL CLUSTERINGS

First, we define distance between hierarchical clusterings. Let $X = \{x_1, \ldots, x_n\}$ be a dataset, $x \in X$ be a data point, and $T$ and $T'$ be hierarchical clusterings over $X$ and $X \setminus \{x\}$, respectively. Then, the distance $d_x(T, T')$ between $T$ and $T'$ is defined recursively as follows. If both $T$ and $T'$ are empty trees, then $d_x(T, T')$ is defined to be zero. Otherwise, we incur the cost of one if

> leaves(left($T$)) $\triangle$ leaves(left($T'$)) $\not\subseteq \{x\}$, or
>
> leaves(right($T$)) $\triangle$ leaves(right($T'$)) $\not\subseteq \{x\}$.

In words, we incur the cost of one if the left subtrees or the right subtrees differ besides the ignored element $x \in X$. Then, we recursively compute the costs $d_x(\mathsf{left}(T), \mathsf{left}(T'))$ and $d_x(\mathsf{right}(T), \mathsf{right}(T'))$ and add them up. The details are given in Algorithm 1. It is easy to verify that $d_x$ satisfies the triangle inequality. Also, note that $d_x(T, T') \leq |T| + |T'|$, where $|T|$ is the number of nodes in $T$ (including the leaves).

---

**Algorithm 1:** Distance between trees

1 **Procedure** $d_x(T, T')$
2    **if** $T = T' = \emptyset$ **then return** 0;
3    **if**
     leaves(left($T$)) $\triangle$ leaves(left($T'$)) $\not\subseteq$ $\{x\}$ *or*
     leaves(right($T$)) $\triangle$ leaves(right($T'$)) $\not\subseteq$ $\{x\}$ **then**
4      $\quad c \leftarrow 1$.
5    **else**
6      $\quad c \leftarrow 0$.
7    **return** $c + d_x(\mathsf{left}(T), \mathsf{left}(T')) + d_x(\mathsf{right}(T), \mathsf{right}(T'))$.

---

### 4.2 AVERAGE SENSITIVITY

Now we define the average sensitivity of a deterministic algorithm as follows:

**Definition 4.1** (Varma and Yoshida (2021)). Let $A$ be a deterministic algorithm that, given a dataset $X = \{x_1, \ldots, x_n\}$, outputs a hierarchical clustering. Then, the average sensitivity of $A$ on a dataset $X = \{x_1, \ldots, x_n\}$ is

$$\frac{1}{n} \sum_{x \in X} d_x(A(X), A(X \setminus \{x\})). \tag{1}$$

To extend the definition to randomized algorithms, we define $\mathrm{EM}_x$ as the earth mover's distance between two distributions with the underlying distance $d_x$. Specifically, for distributions over hierarchical clusterings $\mathcal{T}$ and $\mathcal{T}'$, we define $\mathrm{EM}_x(\mathcal{T}, \mathcal{T}') = \min_{\mathcal{D}} \mathbf{E}_{(\boldsymbol{T}, \boldsymbol{T}') \sim \mathcal{D}} d_x(\boldsymbol{T}, \boldsymbol{T}')$, where $\mathcal{D}$ runs over distributions over pairs of hierarchical clusterings such that the marginal distributions on the first and second coordinates are equal to $\mathcal{T}$ and $\mathcal{T}'$, respectively (sometimes called a *coupling* between $\boldsymbol{T}$ and $\boldsymbol{T}'$ in the literature). Then, we define the average sensitivity of a randomized algorithm as follows:

**Definition 4.2** (Varma and Yoshida (2021)). Let $A$ be a randomized algorithm that, given a dataset $X = \{x_1, \ldots, x_n\}$, outputs a hierarchical clustering. Then, the average sensitivity of $A$ on a dataset $X = \{x_1, \ldots, x_n\}$ is

$$\frac{1}{n} \sum_{x \in X} \mathrm{EM}_x(A(X), A(X \setminus \{x\})).$$

Note that this definition coincides with the one for deterministic algorithms when the algorithm is deterministic.

Sometimes we want to guarantee that a randomized algorithm $A$ outputs similar hierarchical clusterings on $X$ and $X \setminus \{x\}$ when we use the same random coins. For a bit string $\pi$, let $A_\pi$ denote the deterministic algorithm obtained from $A$ by fixing the outcomes of its random coins to $\pi$. Then, we define the following variant of average sensitivity.

**Definition 4.3.** Let $A$ be a randomized algorithm that, given a dataset $X = \{x_1, \ldots, x_n\}$, outputs a hierarchical clustering. Then, the average sensitivity of $A$ under shared randomness on a dataset $X = \{x_1, \ldots, x_n\}$ is

$$\mathbf{E}_\pi \left[ \frac{1}{n} \sum_{x \in X} d_x(A_\pi(X), A_\pi(X \setminus \{x\})) \right].$$

## 5   STABLE-ON-AVERAGE HIERARCHICAL CLUSTERING

### 5.1   ALGORITHM DESCRIPTION

In this section, we describe our algorithm for hierarchical clustering with low average sensitivity, and then derive some theoretical properties. In Section 6, we consider another algorithm with low average sensitivity under shared randomness.

Our algorithm, SHC (Stable Hierarchical Clustering), is given in Algorithm 2. Given a dataset $X = \{x_1, \ldots, x_n\}$ and a parameter $\alpha > 0$, we first transform $X$ into a weighted graph $G = (V, E, w)$, where $V = \{1, 2, \ldots, n\}$, $E = \binom{V}{2}$, and $w(i, j) = \exp(-\alpha \|x_i - x_j\|^2)$, and then pass $G$ to a subroutine REC, which constructs a hierarchical clustering using $G$. Note that closer data point pairs get higher weights in $w$. If $\alpha$ is small, then every data point pair gets almost identical weight, and if $\alpha$ is large, then distant data point pairs get negligible weights and will be ignored in hierarchical clustering.

The subroutine REC is recursive. Given a weighted graph $G = (V, E, w)$ and a depth limit $D \geq 0$, we split the vertex set into two components using a subroutine SSC (Stable Sparse Cut, Algorithm 3), and then recursively process them until the depth reaches $D$.

Now we explain the details of the subroutine SSC. Ideally, we want to solve the *sparsest cut problem*, for which the goal is to compute $S \subseteq V$ that minimizes $\phi_G(S)$. However, approximating $\phi_G(S)$ to within a constant factor is NP-Hard (Chawla et al., 2006), and although some polynomial-time approximation algorithms (Arora et al., 2009; Leighton and Rao,

---

**Algorithm 2:** Stable hierarchical clustering

1 **Procedure** REC($G = (V, E, w), \lambda, d, D$)
2     **if** $d = D$ *or* $|V| = 0$ **then**
3         └ **return** an empty tree.
4     **else if** $|V| = 1$ **then**
5         **return** the only vertex in $G$ as a singleton tree.
6     **else**
7         $\boldsymbol{S} \leftarrow \mathrm{SSC}(G, \lambda)$;
8         $\boldsymbol{T}_1 \leftarrow \mathrm{REC}(G[\boldsymbol{S}], \lambda, d+1)$;
9         $\boldsymbol{T}_2 \leftarrow \mathrm{REC}(G[V \setminus \boldsymbol{S}], \lambda, d+1)$;
10         Let $\boldsymbol{T}$ be the tree with a root node having $\boldsymbol{T}_1$ and $\boldsymbol{T}_2$ as its subtrees;
11         **return** $\boldsymbol{T}$.

12 **Procedure** SHC($X = \{x_1, \ldots, x_n\}, \alpha, \lambda, D$)
13     $V \leftarrow \{1, 2, \ldots, n\}$;
14     $E \leftarrow \binom{V}{2}$;
15     $w(i, j) \leftarrow \exp(-\alpha \|x_i - x_j\|^2)$;
16     $G \leftarrow (V, E, w)$;
17     **return** REC($G, \lambda, 0, D$).

---

**Algorithm 3:** Stable sparsest cut

1 **Procedure** SSC($G = (V, E, w), \lambda$)
2     **for** $\{i, j\} \in \binom{V}{2}$ *with* $i < j$ **do**
3         $S_{ij} \leftarrow \{k \in V : w(i, k) > w(j, k)\}$;
4         └ $\phi_G(i, j) \leftarrow \phi_G(S_{ij})$.
5     Sample a pair $\{\boldsymbol{i}, \boldsymbol{j}\}$ from the distribution that emits $\{i, j\} \in \binom{V}{2}$ with probability $\propto \exp(-\lambda \phi_G(i, j))$;
6     **return** $S_{\boldsymbol{ij}}$.

---

1999) are known, they are slow in practice because they internally solve LPs or SDPs, and it is not clear whether these are stable. Hence, we take a different approach.

Our idea is to select a pair of vertices, called *centroids*, and then assign every other vertex to the more similar centroid to form a partition into two components. To achieve a small average sensitivity, we select the pair of centroids as follows. For $\{i, j\} \in \binom{V}{2}$ with $i < j$, let $S_{ij} = \{k \in V : w(i, k) > w(j, k)\}$ be the set of vertices that is more similar to $i$ than $j$, and define $\phi_G(i, j) = \phi_G(S_{ij})$. Then, we sample a pair of centroids $\{\boldsymbol{i}, \boldsymbol{j}\}$ using the exponential mechanism with the cost function $\phi_G(\cdot, \cdot)$ and the given parameter $\lambda$. When $\lambda = 0$, the exponential mechanism returns $\{\boldsymbol{i}, \boldsymbol{j}\}$ uniformly sampled from $\binom{V}{2}$, and when $\lambda = \infty$, it returns $\{i, j\}$ that minimizes $\phi_G(i, j)$.

## 5.2 THEORETICAL PROPERTIES

The time complexity of SHC is easy to analyze:

**Theorem 5.1.** *The time complexity of* SHC *is* $O(Dn^3)$.

Next, we discuss the approximation guarantee and (a variant of) the average sensitivity of SSC. For a weighted graph $G = (V, E, w)$, we define

$$\phi_G^* = \min_{\{i,j\} \in \binom{V}{2}} \phi_G(S_{ij}).$$

Note that $\phi_G^*$ is *not* the minimum sparsity of a set in $G$, i.e., $\min_{S \subseteq V} \phi_G(S)$. Let $w_G$ denote the total edge weight, that is, $\sum_{\{i,j\} \in \binom{V}{2}} w(i, j)$. The following holds:

**Theorem 5.2.** *For a weighted graph $G$ of $n$ vertices and $\lambda > 0$, let $\boldsymbol{S} = \mathrm{SSC}(G, \lambda)$. Then, we have*

$$\mathbf{E}[\phi_G(\boldsymbol{S})] \leq \phi_G^* + O\left(\frac{\log(\lambda w_G)}{\lambda}\right).$$

*We also have*

$$\frac{1}{n} \sum_{k \in V} d_{\mathrm{TV}}(\mathrm{SSC}(G, \lambda), \mathrm{SSC}(G - k, \lambda)) = O\left(\frac{1}{n}\left(\lambda \phi_G^* + \log(n w_G)\right)\right).$$

Because the weight function $w$ is $[0, 1]$-valued, $\phi_G^* = O(1)$ and $w_G = O(n^2)$. Then for $\epsilon > 0$, we obtain $\mathbf{E}[\phi_G(\boldsymbol{S})] = (1 + \epsilon)\phi_G*$ by setting $\lambda = \Theta(\log n / (\epsilon \phi_G^*))$. For this particular choice of $\lambda$, the average total variation distance is $O(\log n / (\epsilon n))$, which is quite small.

Finally, we discuss the average sensitivity of SHC.

**Theorem 5.3.** *The average sensitivity of* SHC*($X, \alpha, \lambda, D$) is* $O\left(D(\lambda w_G/n + \log(n w_G))\right)$*, where $G$ is the graph constructed by using $X$ and $\alpha$ in* SHC.

Recalling that $w_G = O(n^2)$, the bound is roughly $O(\lambda Dn)$, which can be made small by setting $\lambda \ll 1$.

# 6 STABLE-ON-AVERAGE HIERARCHICAL CLUSTERING UNDER SHARED RANDOMNESS

In this section, we propose an algorithm SHC-SR by modifying SHC (Algorithm 2) so that it has a small average sensitivity under shared randomness.

First, we design a randomized algorithm called SAMPLING that, given a vector $p \in \mathbb{R}_+^n$ with $\sum_{i=1}^n p_i = 1$, and a random bit string $\boldsymbol{\pi}$, outputs $i \in \{1, \ldots, n\}$ with probability $p_i$ such that perturbing the vector $p$ does not change the output with high probability over $\boldsymbol{\pi}$.

For a set $S$, let $U(S, \pi)$ denote a procedure that outputs an element $i \in S$ such that $U(S, \boldsymbol{\pi})$ for a random bit string $\boldsymbol{\pi}$ provides a uniform distribution over $S$. Such a procedure can be easily implemented by taking the first few bits from $\boldsymbol{\pi}$ and then map them to an element in $S$. Then in SAMPLING($p, \pi$), we first compute a permutation $\sigma$ so that $p_{\sigma(1)} \leq \cdots \leq p_{\sigma(n)}$ and compute some carefully designed vector $q \in [0, 1]^n$ using $p$ and $\sigma$. Then, we sample $t \in [0, 1]$ and $i \in [0, 1]$ uniformly at random and if $q_i > t$, then we return $i$, and otherwise, we repeat the process. The vector $q$ is designed so that this process outputs $i$ with probability $p_i$. The details are given in Algorithm 4.

Because the only randomized process in SHC is the exponential mechanism used in SSC (Algorithm 3), by replacing it with SAMPLING that simulates the exponential mechanism, we obtain a hierarchical clustering algorithm SHC-SR with low average sensitivity under shared randomness:

**Theorem 6.1.** *There exists an algorithm* SHC-SR *that, given a dataset* $X = (x_1, \ldots, x_n)$*,* $\alpha \geq 0$*,* $\lambda \geq 0$*, an integer* $D$*, and a bit string* $\pi$*, outputs a hierarchical clustering over* $X$ *such that*

- *the distribution of* SHC-SR$(X, \alpha, \lambda, D, \pi)$ *over random bits* $\pi$ *is equal to that of* SHC$(X, \alpha, \lambda, D)$*.*

- *the average sensitivity of* SHC-SR$(X, \alpha, \lambda, D, \pi)$ *under shared randomness is* $O\left(D(\lambda w_G/n + \log(n w_G))\right)$*, where* $G$ *is the graph constructed by using* $X$ *and* $\alpha$ *as in* SHC*.*

---

**Algorithm 4:** Sampling with a low average sensitivity under shared randomness

1 **Procedure** SAMPLING$(p, \pi)$
2      Let $\sigma$ be a permutation such that $p_{\sigma(1)} \leq p_{\sigma(2)} \leq \ldots \leq p_{\sigma(n)}$;
3      Let $q \in \mathbb{R}_+^n$ so that $q_{\sigma(i)} = q_{\sigma(i-1)} + (n - i + 1)(p_{\sigma(i)} - p_{\sigma(i-1)})$, where $p_0 = q_{\sigma(0)} = 0$;
4      $t \leftarrow U([0, 1], \pi)$ and delete the used bits from $\pi$;
5      **while true do**
6          $i \leftarrow U(\{1, 2, \ldots, n\}, \pi)$ and delete the used bits from $\pi$;
7          **if** $q_i > t$ **then**
8              **break**.
9          **return** $i$.

---

# 7 EXPERIMENTS

We demonstrate that the proposed SHC-SR (Section 6) can output stable hierarchical clustering using some benchmark datasets. For all the experiments, we used a workstation with 48 cores of AMD EPYC processors and 256GB of RAMS.

## 7.1 SETUPS

**Datasets** We took three datasets shown in Table 1 from `sklearn.datasets`. For the experiments, we subsampled a fraction of the data points from a dataset so that we can assess the effect of the data size $n$.

Table 1: Datasets

| dataset | data size | # of features |
|---|---|---|
| breast cancer | 569 | 30 |
| diabetes | 442 | 10 |
| digits | 1797 | 64 |

**Hierarchical Clustering Algorithms** In the experiment, we implemented SHC-SR given in Theorem 6.1. We constructed weighted graphs by setting $w(i, j) = \exp(-\alpha \|x_i - x_j\|^2/m)$ with $m$ being the median of all the pairwise distance, and varied $\alpha$ to some different values. We also varied the parameter $\lambda$ used in SSC-SR. The case $\lambda = \infty$ corresponds to a greedy algorithm that selects the pair $(i, j)$ with the smallest $\phi_G(i, j)$ in SSC-SR (Algorithm 3 with the exponential mechanism being implemented with SAMPLING), and the case $\lambda = 0$ corresponds to an algorithm that selects the pair $(i, j)$ uniformly at random in SSC-SR. We implemented SHC-SR in Python 3 using the JIT compiler of Numba.

We adopted some standard hierarchical clustering algorithms as baseline methods for comparison.[1] As typical agglomerative clustering algorithms, we adopted four algorithms implemented in `AgglomerativeClustering` in scikit-learn with four different linkage criterion: `ward`, `average`, `complete`, and `single`, with the other options set to default. We note that Balcan et al. (2014) reported that `ward` tends to be robust against outlier injections and noise contamination. As the representatives of divisive clustering, we adopted bisecting 2-means (Jain, 2010) and principal direction divisive partitioning (Boley, 1998). These two methods recursively split data points by using the standard 2-means clustering and the sign of the first principal component, respectively. We implemented these methods, which we denote by `2-means` and `pcdd`, by using `KMeans` in scikit-learn with the number of clusters set to two and ten random initializations and `PCA` with the number of components set to one, respectively, and default parameters for the other options.

---

[1] We did not adopt the outlier–robust methods (Eriksson et al., 2011; Balcan et al., 2014; Cheng et al., 2019) because the core of these methods is on identifying outliers, which is irrelevant to the current problem.

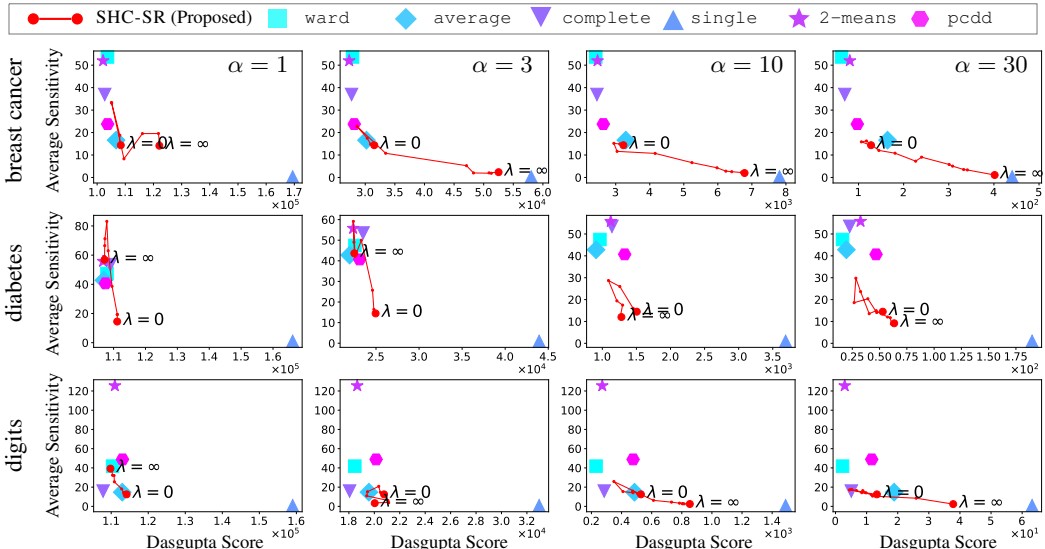

Figure 2: Trade-offs between average sensitivity and Dasgupta score for the data size $n = 100$, depth $D = 10$, and $\alpha = 1, 3, 10, 30$. The results of SHC-SR are shown in red lines displaying the results for several different $\lambda \in \{0, 10^{-3}, \ldots, 10^6, \infty\}$.

**Evaluation criteria** We measure the average sensitivity of hierarchical clustering algorithms as well as their qualities. We evaluated the average sensitivity following (1). For SHC-SR, we treated SHC-SR$(\cdot, \alpha, \lambda, D, \pi)$ with a fixed $\pi$ as the deterministic algorithm $A$.

As the quality measure, we adopted two popular criteria, *Dasgupta score* (Dasgupta, 2016), *Dendrogram purity* (Heller and Ghahramani, 2005), and *Cophenetic Correlation* (Sokal and Rohlf, 1962) [2]. Dasgupta score measures the quality of a hierarchical clustering $T$ using costs of pairs of data points. More specifically, we define the Dasgupta score of a hierarchical clustering $T$ by $\mathrm{score}(T) = \sum_{i,j=1; i \neq j}^{n} w(i,j)n(i,j)$, where $n(i,j)$ denotes the number of data points belonging to the subtree rooted at the lowest common ancestor of nodes that $x_i$ and $x_j$ belong to. The Dasgupta score is small when dissimilar points $x_i$ and $x_j$ (i.e., $w(i,j)$ is small) are split into different clusters in a shallow part of the tree, and similar points (i.e., $w(i,j)$ is large) are split in a deeper part. Thus, a clustering $T$ with smaller $\mathrm{score}(T)$ is considered ideal.

**Procedure** We generated 10 subsampled datasets of size $n = 100, 300$, and $500$ from the original dataset. [3] For each subsampled dataset, we constructed a hierarchical clustering using SHC-SR over different values of $\lambda$ and the baseline methods. As the result, we obtained 10 clusterings for each method. We compute the average of the average sensitivity and the Dasgupta score of these 10 clusterings. We then report the trade-offs of the average of the average sensitivity and the average of the clustering qualities.

### 7.2 RESULTS

Figures 2 shows the results of the experiments with $n = 100$. Each figure shows the trade-offs between the average sensitivity and the average Dasgupta score, with the depth of $T$ limited to 10, and with the similarity coefficient $\alpha$ varied to 1, 3, 10, and 30. The results of the baselines and SHC-SR for several different $\lambda$ are shown in different symbols and red lines, respectively. We can find that the red lines of SHC-SR tend to lie in the lower left area of the figures. That is, SHC-SR with appropriately chosen $\lambda$ can attain a good trade-off with small average sensitivity and better Dasgupta scores, as expected. By contrast, all the baselines, except for `single`, tend to exhibit small Dasgupta scores while incurring high average sensitivity. These methods are therefore good at producing high quality clusterings, while being sensitive to a small perturbation of the dataset. The

---

[2] We show the results for Dendrogram purity and Cophenetic Correlation in Appendix.

[3] We show the results for $n = 300$ and $n = 500$ in Appendix because they are similar to $n = 100$.

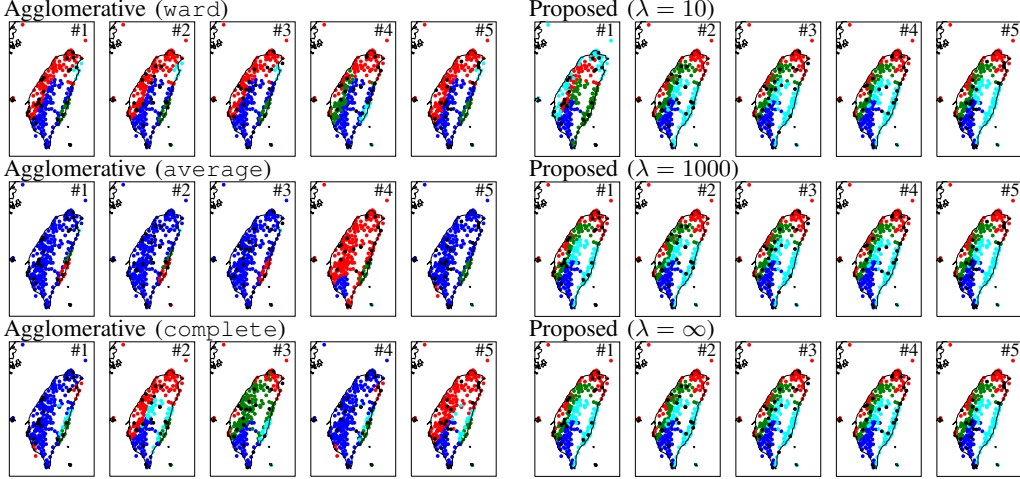

Figure 3: Clustering results on a GPS dataset over the five trials with 20 data points removed. The figures show four clusters at the depth two of the obtained hierarchy. The colored dots in blue, red, green, and cyan indicate data points within the same clusters. The black dots denote removed data points. We used $\alpha = 0.1$ for the proposed method.

result of `single` is exceptional, exhibiting large Dasgupta scores with small average sensitivity. We observed that `single` tends to produce highly unbalanced clusterings because `single` split the dataset into small and large clusters. Although such a split is less sensitive to the dataset perturbation and has smaller average sensitivity, the quality of clustering is poor. SHC-SR provides a way to balance the quality of the clustering and its average sensitivity by tuning $\lambda$ upon the user demand.

## 8 APPLICATION TO GPS DATASET

We applied SHC-SR and agglomerative algorithms to a real-world problem involving a GPS dataset (Takahashi et al., 2019).[4] This dataset consists of 280 GPS markers in Taiwan, where each data point represents its longitude, latitude, and velocity in the horizontal directions. By applying clustering to the horizontal velocities, we can cluster regions with similar movements and find active tectonic boundaries. The stability of clustering is crucial in this application because if the found clusters change drastically upon removal of a few GPS markers, the clusters may be an artifact induced by unstable clustering algorithms rather than the true tectonic boundaries.

Figure 3 shows the clustering results on the GPS dataset over the five trials when randomly chosen 20 out of 280 points are removed from the dataset. Here, we display the four clusters found at the depth two of the obtained hierarchy. The figures show that the agglomerative algorithms (`ward`, `average`, `complete`) tend to produce different clusters over different data removals. By contrast, SHC-SR with $\lambda = 10, 1000$, and $\infty$ produce almost identical clusters, except the first result on $\lambda = 10$. This result confirms that we can obtain stable clusters by using SHC-SR.

## 9 CONCLUSIONS

In this work, we considered the average sensitivity of hierarchical clustering. We proposed hierarchical clustering algorithms SHC and SHC-SR and theoretically proved that they have low average sensitivity and average sensitivity under shared randomness, respectively. Then using real-world datasets, we empirically confirmed that our algorithm SHC-SR achieves a good trade-off between the quality of the output clustering and average sensitivity.

---

[4]We omitted `single`, `2-means`, and `pcdd` because of their poor performances in the previous experiments; `single` was poor at its clustering quality, `2-means` was poor at its average sensitivity, and `pcdd` tends to be parteo-dominated by other methods.

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

## A  PROOF OF THEOREM 5.1

*Proof of Theorem 5.1.* Note that SSC on a graph of $k$ nodes takes $O(k^3)$ time. Let $\boldsymbol{n}_{d,i}$ be the size of the graph in the $i$-th call of REC at depth $d$. Note that we have $2^d$ calls at depth $d$, and hence $1 \leq i \leq 2^d$. Also note that $\sum_{i=1}^{2^d} \boldsymbol{n}_{d,i} = n$. Then, the total running time is

$$
O\left(\sum_{d=0}^{D}\sum_{i=1}^{2^d} \boldsymbol{n}_{d,i}^3\right) = O\left(\sum_{d=0}^{D} n^3\right) = O\left(Dn^3\right). \qquad \square
$$

## B  PROOF OF THEOREM 5.2

We analyze approximation guarantee and average total variation distance of SSC in Sections B.1 and B.2, respectively.

### B.1  APPROXIMATION GUARANTEE

The following lemma shows the approximation guarantee part of Theorem 5.2.

**Lemma B.1.** *Let $G$ be a weighted graph, $\lambda > 0$, and $\boldsymbol{S} = \mathrm{SSC}(G, \lambda)$. Then, we have*

$$
\mathbf{E}[\phi_G(\boldsymbol{S})] \leq \phi_G^* + O\left(\frac{\log(\lambda w_G)}{\lambda}\right).
$$

*Proof.* Note that $\phi_G(i,j) \leq w_G/n$. Then by Lemma 3.1, we have

$$
\mathbf{E}[\phi_G(\boldsymbol{S})] = \phi_G^* + \frac{\log\binom{n}{2}}{\lambda} + \frac{t}{\lambda} + e^{-t} \cdot \frac{w_G}{n}.
$$

By setting $t = \log(\lambda w_G/n)$, we have

$$
\begin{aligned}
\mathbf{E}[\phi_G(\boldsymbol{S})] &= \phi_G^* + \frac{\log\binom{n}{2}}{\lambda} + \frac{\log(\lambda w_G/n)}{\lambda} + \frac{1}{\lambda} \\
&= \phi_G^* + O\left(\frac{\log(\lambda w_G)}{\lambda}\right),
\end{aligned}
$$

as desired. $\qquad \square$

### B.2  AVERAGE TOTAL VARIATION DISTANCE

In this section, we analyze the average total variation distance of SSC. Throughout this section, we fix the input weighted graph $G = (V, E, w)$ and $\lambda > 0$.

Let $Z = \sum_{\{i,j\}\in\binom{V}{2}} \exp(-\lambda\phi_G(i,j))$ and $Z^{(k)} = \sum_{\{i,j\}\in\binom{V^{-k}}{2}} \exp(-\lambda\phi_{G-k}(i,j))$. We note that, for graphs $G$ and $G-k$, we sample a pair $(i,j)$ with probability $p(i,j) := \exp(-\lambda\phi_G(i,j))/Z$ and $p^{(k)}(i,j) := \exp(-\lambda\phi_G(i,j))/Z^{(k)}$, respectively. The following techinical lemma is useful to bound the total variation distance between the distributions for $G$ and $G-k$. We postpone the proof to Section B.3.

**Lemma B.2.** *We have*

$$
\begin{aligned}
\frac{1}{Z}\sum_{k\in V}\sum_{\{i,j\}\in\binom{V^{-k}}{2}} &|e^{-\lambda\phi_G(i,j)} - e^{-\lambda\phi_{G-k}(i,j)}| \\
&= O\left(\lambda\phi_G^* + \log(nw_G)\right).
\end{aligned}
$$

*Note that this also implies $\sum_{k\in V} |Z - Z^{(k)}|/Z = O\left(\lambda\phi_G^* + \log(nw_G)\right)$.*

The following shows the average total variation distance part of Theorem 5.2.

**Lemma B.3.** *Let $\boldsymbol{S} = \mathrm{SSC}(G, \lambda)$ and $\boldsymbol{S}^{(k)} = \mathrm{SSC}(G - k, \lambda)$. Then, we have*

$$\sum_{k \in V} d_{\mathrm{TV}}(\boldsymbol{S}, \boldsymbol{S}^{(k)}) = O\left(\lambda \phi_G^* + \log(n w_G)\right).$$

*Proof.* Note that $\boldsymbol{S}$ is deterministically constructed from the pair of centroids $\{\boldsymbol{i}, \boldsymbol{j}\}$. Hence, we consider the change of the pair $\{\boldsymbol{i}, \boldsymbol{j}\}$ in total variation distance between $G$ and $G - k$. First, we have

$$\sum_{k \in V} \sum_{\{i,j\} \in \binom{V^{-k}}{2}} |p(i,j) - p^{(k)}(i,j)|$$

$$= \sum_{k \in V} \sum_{\{i,j\} \in \binom{V^{-k}}{2}} \left| \frac{e^{-\lambda \phi_G(i,j)}}{Z} - \frac{e^{-\lambda \phi_{G-k}(i,j)}}{Z^{(k)}} \right|$$

$$= \sum_{k \in V} \sum_{\{i,j\} \in \binom{V^{-k}}{2}} \left| \frac{e^{-\lambda \phi_G(i,j)}}{Z} - \frac{e^{-\lambda \phi_{G-k}(i,j)}}{Z} \cdot \left(1 + \frac{Z - Z^{(k)}}{Z^{(k)}}\right) \right|$$

$$\leq \frac{1}{Z} \sum_{k \in V} \sum_{\{i,j\} \in \binom{V^{-k}}{2}} |e^{-\lambda \phi_G(i,j)} - e^{-\lambda \phi_{G-k}(i,j)}|$$

$$\quad + \sum_{k \in V} \frac{|Z - Z^{(k)}|}{ZZ^{(k)}} \sum_{\{i,j\} \in \binom{V^{-k}}{2}} e^{-\lambda c_k(i,j)}$$

$$= \frac{1}{Z} \sum_{k \in V} \sum_{\{i,j\} \in \binom{V^{-k}}{2}} |e^{-\lambda \phi_G(i,j)} - e^{-\lambda \phi_{G-k}(i,j)}| + \sum_{k \in V} \frac{|Z - Z^{(k)}|}{Z}$$

$$= O\left(\lambda \phi_G^* + \log(n w_G)\right). \qquad \text{(by Lemma B.2)}$$

Then, we have

$$\sum_{k \in V} d_{\mathrm{TV}}(\boldsymbol{S}, \boldsymbol{S}^{(k)})$$

$$\leq \sum_{k \in V} \left( \sum_{\{i,j\} \in \binom{V^{-k}}{2}} |p(i,j) - p^{(k)}(i,j)| + \sum_{\{i,j\} \in \binom{V}{2} \setminus \binom{V^{-k}}{2}} p(i,j) \right)$$

$$= O\left(\lambda \phi_G^* + \log(n w_G)\right) + \sum_{\{i,j\} \in \binom{V}{2}} p(i,j)$$

$$= O\left(\lambda \phi_G^* + \log(n w_G)\right). \qquad \square$$

Theorem 5.2 follows by combining Lemma B.1 and Lemma B.3.

### B.3 PROOF OF LEMMA B.2

For $k \in V \setminus \{i, j\}$, we write $S_{ij}^{(k)}$ to denote $S_{ij}$ computed for the instance $G - k$. We write $\bar{S}_{ij}$ to denote $V \setminus S_{ij}$. We start with the following simple lemma.

**Lemma B.4.** *For $\{i, j\} \in \binom{V}{2}$ with $i < j$ and $k \in V \setminus \{i, j\}$,*

$$\phi_G(i,j) - \phi_{G-k}(i,j)$$

$$= \begin{cases} \dfrac{c_G(S_{ij}, \{k\}) \cdot |\bar{S}_{ij}| - c_G(S_{ij}, \bar{S}_{ij})}{|S_{ij}| \cdot |\bar{S}_{ij}| \cdot (|\bar{S}_{ij}| - 1)} & \text{if } k \in \bar{S}_{ij}, \\[3mm] \dfrac{c_G(\{k\}, \bar{S}_{ij}) \cdot |S_{ij}| - c_G(S_{ij}, \bar{S}_{ij})}{(|S_{ij}| - 1) \cdot |S_{ij}| \cdot |\bar{S}_{ij}|} & \text{if } k \in S_{ij}. \end{cases}$$

*Proof.* For notational simplicity, we write $S$, $\bar{S}$, and $S^{(k)}$ to denote $S_{ij}$, $\bar{S}_{ij}$, and $S_{ij}^{(k)}$, respectively.

If $k \in \bar{S}$, then $S = S^{(k)}$ and we have

$$\phi_G(i, j) - \phi_{G-k}(i, j)$$

$$= \frac{c_G(S, V \setminus S)}{|S| \cdot |V \setminus S|} - \frac{c_{G-k}(S^{(k)}, (V - k) \setminus S^{(k)})}{|S^{(k)}| \cdot |(V - k) \setminus S^{(k)}|}$$

$$= \frac{c_G(S, V \setminus S)}{|S| \cdot |V \setminus S|} - \frac{c_G(S, V \setminus S) - c_G(S, \{k\})}{|S| \cdot (|V \setminus S| - 1)}$$

$$= \frac{c_G(S, \bar{S})(|\bar{S}| - 1) - (c_G(S, \bar{S}) - c_G(S, \{k\})) \cdot |\bar{S}|}{|S| \cdot |\bar{S}| \cdot (|\bar{S}| - 1)}$$

$$= \frac{c_G(S, \{k\}) \cdot |\bar{S}| - c_G(S, \bar{S})}{|S| \cdot |\bar{S}| \cdot (|\bar{S}| - 1)}.$$

Otherwise, that is, if $k \in S$, then $S = S^{(k)} \cup \{k\}$ and we have

$$\phi_G(S) - \phi_{G-k}(S^{(k)})$$

$$= \frac{c_G(S, V \setminus S)}{|S| \cdot |V \setminus S|} - \frac{c_{G-k}(S^{(k)}, (V - k) \setminus S^{(k)})}{|S^{(k)}| \cdot |(V - k) \setminus S^{(k)}|}$$

$$= \frac{c_G(S, V \setminus S)}{|S| \cdot |V \setminus S|} - \frac{c_G(S, V \setminus S) - c_G(\{k\}, V \setminus S)}{(|S| - 1) \cdot |V \setminus S|}$$

$$= \frac{c_G(S, \bar{S}) \cdot (|S| - 1) - (c_G(S, \bar{S}) - c_G(\{k\}, \bar{S})) \cdot |S|}{(|S| - 1) \cdot |S| \cdot |\bar{S}|}$$

$$= \frac{c_G(\{k\}, \bar{S}) \cdot |S| - c_G(S, \bar{S})}{(|S| - 1) \cdot |S| \cdot |\bar{S}|}. \qquad \square$$

Next, we bound the total change of the sparsity of the set induced by a fixed pair of centroids over deleted vertices.

**Lemma B.5.** *For any* $\{i, j\} \in \binom{V}{2}$ *with* $i < j$, *we have*

$$\sum_{k \in V \setminus \{i, j\}} |\phi_G(i, j) - \phi_{G-k}(i, j)| = O(\phi_G(i, j)).$$

*Proof.* For notational simplicity, we write $S$, $\bar{S}$, and $S^{(k)}$ to denote $S_{ij}$, $\bar{S}_{ij}$, and $S_{ij}^{(k)}$, respectively.

Let $A = \{k \in V \setminus \{i, j\} : \phi_G(S) \le \phi_{G-k}(S^{(k)})\}$. Then, we have

$$\sum_{k \in A} |\phi_G(S) - \phi_{G-k}(S^{(k)})| = \sum_{k \in A} (\phi_{G-k}(S^{(k)}) - \phi_G(S))$$

$$= \sum_{k \in A \cap \bar{S}} \frac{c_G(S, \bar{S}) - c_G(S, \{k\}) \cdot |\bar{S}|}{|S| \cdot |\bar{S}| \cdot (|\bar{S}| - 1)}$$

$$+ \sum_{k \in A \cap S} \frac{c_G(S, \bar{S}) - c_G(\{k\}, \bar{S}) \cdot |S|}{(|S| - 1) \cdot |S| \cdot |\bar{S}|} \qquad \text{(by Lemma B.4)}$$

$$\le \sum_{k \in A \cap \bar{S}} \frac{c_G(S, \bar{S})}{|S| \cdot |\bar{S}| \cdot (|\bar{S}| - 1)} + \sum_{k \in A \cap S} \frac{c_G(S, \bar{S})}{(|S| - 1) \cdot |S| \cdot |\bar{S}|}$$

$$= \phi_G(S) \cdot \left( \sum_{k \in A \cap \bar{S}} \frac{1}{|\bar{S}| - 1} + \sum_{k \in A \cap S} \frac{1}{|S| - 1} \right) = O(\phi_G(S)).$$

Let $\bar{A} = (V \setminus \{i, j\}) \setminus A$. Then, we have

$$\sum_{k \in \bar{A}} |\phi_G(S) - \phi_{G-k}(S_k)| = \sum_{k \in \bar{A}} (\phi_G(S) - \phi_{G-k}(S_k))$$

$$
\begin{aligned}
&= \sum_{k \in \bar{A} \cap \bar{S}} \frac{c_G(S, \{k\}) \cdot |\bar{S}| - c_G(S, \bar{S})}{|S| \cdot |\bar{S}| \cdot (|\bar{S}| - 1)} \\
&\quad + \sum_{k \in \bar{A} \cap S} \frac{c_G(\{k\}, \bar{S}) \cdot |S| - c_G(S, \bar{S})}{(|S| - 1) \cdot |S| \cdot |\bar{S}|} \qquad \text{(by Lemma B.4)} \\
&\le \sum_{k \in \bar{A} \cap \bar{S}} \frac{c_G(S, \{k\})}{|S| \cdot (|\bar{S}| - 1)} + \sum_{k \in \bar{A} \cap S} \frac{c_G(\{k\}, \bar{S})}{(|S| - 1) \cdot |\bar{S}|} \\
&= O\left( \frac{c_G(S, \bar{A} \cap \bar{S})}{|S| \cdot |\bar{S}|} \right) + O\left( \frac{c_G(\bar{A} \cap S, \bar{S})}{|S| \cdot |\bar{S}|} \right) = O(\phi_G(S)).
\end{aligned}
$$

Combining the above two, we have

$$
\begin{aligned}
&\sum_{k \in V \setminus \{i,j\}} |\phi_G(S) - \phi_{G-k}(S_k)| \\
&= \sum_{k \in A} |\phi_G(S) - \phi_{G-k}(S^{(k)})| + \sum_{k \in \bar{A}} |\phi_G(S) - \phi_{G-k}(S^{(k)})| \\
&= O(\phi_G(S)). \qquad\qquad\qquad\qquad\qquad\qquad\qquad\qquad \square
\end{aligned}
$$

*Proof of Lemma B.2.* Let $A_k = \{\{i,j\} \in \binom{V^{-k}}{2} \mid \phi_G(i,j) \le \phi_{G-k}(i,j)\}$. Then, we have

$$
\begin{aligned}
&\sum_{\{i,j\} \in \binom{V^{-k}}{2}} |e^{-\lambda \phi_G(i,j)} - e^{-\lambda \phi_{G-k}(i,j)}| \\
&= \sum_{\{i,j\} \in A} \left( e^{-\lambda \phi_G(i,j)} - e^{-\lambda \phi_{G-k}(i,j)} \right) \\
&= \sum_{\{i,j\} \in A} \left( e^{-\lambda \phi_G(i,j)} \big(1 - e^{-\lambda(\phi_{G-k}(i,j) - \phi_G(i,j))}\big) \right) \\
&\le \lambda \sum_{\{i,j\} \in A_k} e^{-\lambda \phi_G(i,j)} (\phi_{G-k}(i,j) - \phi_G(i,j)). \qquad (1 - e^{-x} \le x)
\end{aligned}
$$

Let $\bar{A}_k = \binom{V}{2} \setminus A_k$. Note that we have $\phi_G(i,j) > \phi_{G-k}(i,j)$ for any $\{i,j\} \in \bar{A}_k$. Then, we have

$$
\begin{aligned}
&\sum_{\{i,j\} \in \binom{V^{-k}}{2}} |e^{-\lambda \phi_G(i,j)} - e^{-\lambda \phi_{G-k}(i,j)}| \\
&\le \sum_{\{i,j\} \in \bar{A}_k} \left( e^{-\lambda \phi_{G-k}(i,j)} - e^{-\lambda \phi_G(i,j)} \right) \\
&\le \sum_{\{i,j\} \in \bar{A}_k} \left( e^{-\lambda \phi_G(i,j)} \big( e^{-\lambda(\phi_{G-k}(i,j) - \phi_G(i,j))} - 1 \big) \right) \\
&\le (e-1)\lambda \sum_{\{i,j\} \in \bar{A}_k} e^{-\lambda \phi_G(i,j)} (\phi_G(i,j) - \phi_{G-k}(i,j)). \quad (e^x - 1 \le (e-1)x \text{ for } x \in [0,1])
\end{aligned}
$$

Then, we have

$$
\begin{aligned}
&\frac{1}{Z} \sum_{k \in V} \sum_{\{i,j\} \in \binom{V^{-k}}{2}} |e^{-\lambda \phi_G(i,j)} - e^{-\lambda \phi_{G-k}(i,j)}| \\
&\le \frac{(e-1)\lambda}{Z} \times \\
&\qquad \sum_{k \in V} \sum_{\{i,j\} \in \binom{V^{-k}}{2}} \left( e^{-\lambda \phi_G(i,j)} |\phi_G(i,j) - \phi_{G-k}(i,j)| \right) \\
&= \frac{(e-1)\lambda}{Z} \times
\end{aligned}
$$

$$\sum_{\{i,j\}\in\binom{V}{2}} \left(e^{-\lambda\phi_G(i,j)} \sum_{k\in V\setminus\{i,j\}} |\phi_G(i,j) - \phi_{G-k}(i,j)|\right)$$

$$\leq \frac{(e-1)\lambda}{Z} \sum_{\{i,j\}\in\binom{V}{2}} e^{-\lambda\phi_G(i,j)}\phi_G(i,j) \qquad \text{(by Lemma B.5)}$$

$$= O\left(\lambda\left(\phi_G^* + \frac{\log(nw_G)}{\lambda}\right)\right) \qquad \text{(by Lemma B.1)}$$

$$= O\left(\lambda\phi_G^* + \log(nw_G)\right). \qquad \qquad \square$$

## C  PROOF OF THEOREM 5.3

*Proof of Theorem 5.3.* For notational simplicity, we drop the arguments $\lambda$ and $D$ from $\text{REC}(G,\lambda,d,D)$ because they are fixed in this proof. Also we drop $d$ when it is clear from the context.

For a subgraph $H$ of $G$ and $k \in V := \{1,2,\ldots,n\}$, let $H^{(k)} := H - k$. For $d \geq 0$, let $\boldsymbol{H}_{d,1},\ldots,\boldsymbol{H}_{d,2^d}$ be the random graphs on which REC is called at depth $d$ (if the number of subgraphs on which REC is called at depth $d$ is less than $2^d$, we append empty graphs). We can order them so that $\text{REC}(\boldsymbol{H}_{d,j},d)$ calls $\text{REC}(\boldsymbol{H}_{d+1,2j-1},d+1)$ and $\text{REC}(\boldsymbol{H}_{d+1,2j},d+1)$ (if $\text{REC}(\boldsymbol{H}_{d,j},d)$ does not make recursive calls, we set $\boldsymbol{H}_{d+1,2j-1}$ and $\boldsymbol{H}_{d+1,2j}$ to be empty graphs).

We now show that

$$\mathop{\mathbf{E}}_{\boldsymbol{H}_{d,j}} \sum_{k=1}^{n} \sum_{j=1}^{2^d} \text{EM}_k(\text{REC}(\boldsymbol{H}_{d,j}), \text{REC}(\boldsymbol{H}_{d,j}^{(k)}))$$

$$= O\left((D-d)(\lambda w_G + n\log(nw_G))\right) \qquad (2)$$

by (backward) induction on $d$. Then, the claim holds by setting $d = 0$ and noting that the average sensitivity of $\text{REC}(G,\lambda,0,D)$ is equal to that of $\text{SHC}(X,\alpha,\lambda)$.

For the base case $d = D$, we have

$$\mathop{\mathbf{E}}_{\boldsymbol{H}_{d,j}} \sum_{k=1}^{n} \sum_{j=1}^{2^D} \text{EM}_k(\text{REC}(\boldsymbol{H}_{d,j},d), \text{REC}(\boldsymbol{H}_{d,j}^{(k)},d)) = 0.$$

Let $d < D$ and assume that Hypothesis (2) holds for higher depth. Let $\boldsymbol{S}_{d,j}$ and $\boldsymbol{S}_{d,j}^{(k)}$ be the (random) sets $\boldsymbol{S}$ constructed in $\text{REC}(\boldsymbol{H}_{d,j})$ and $\text{REC}(\boldsymbol{H}_{d,j}^{(k)})$, respectively. For a set $S$, Let $\boldsymbol{H}_{d+1,2j-1}^{S}$ and $\boldsymbol{H}_{d+1,2j}^{S}$ be the two sets obtained by partitioning $\boldsymbol{H}_{d,j}$ according to $S$. Then, we have

$$\sum_{k=1}^{n} \sum_{j=1}^{2^d} \text{EM}_k(\text{REC}(\boldsymbol{H}_{d,j}), \text{REC}(\boldsymbol{H}_{d,j}^{(k)}))$$

$$\leq \sum_{k=1}^{n} \sum_{j=1}^{2^d} \Big(d_{\text{TV}}(\boldsymbol{S}_{d,j}, \boldsymbol{S}_{d,j}^{(k)}) \cdot |V(\boldsymbol{H}_{d,j})| +$$

$$\mathop{\mathbf{E}}_{\boldsymbol{S}_{d,j}} \text{EM}_k(\text{REC}(\boldsymbol{H}_{d+1,2j-1}^{\boldsymbol{S}_{d,j}}), \text{REC}(\boldsymbol{H}_{d+1,2j-1}^{\boldsymbol{S}_{d,j}^{(k)}})) +$$

$$\mathop{\mathbf{E}}_{\boldsymbol{S}_{d,j}} \text{EM}_k(\text{REC}(\boldsymbol{H}_{d+1,2j}^{\boldsymbol{S}_{d,j}}), \text{REC}(\boldsymbol{H}_{d+1,2j}^{\boldsymbol{S}_{d,j}^{(k)}}))\Big)$$

$$\leq \sum_{j=1}^{2^d} O(\lambda\phi_{\boldsymbol{H}_{d,j}}^* + \log(nw_G)) \cdot |V(\boldsymbol{H}_{d,j})| +$$

$$\sum_{k=1}^{n} \sum_{j=1}^{2^{d+1}} \mathop{\mathbf{E}}_{\boldsymbol{S}_{d,j}} \text{EM}_k(\text{REC}(\boldsymbol{H}_{d+1,j}^{\boldsymbol{S}_{d,j}}), \text{REC}(\boldsymbol{H}_{d+1,j}^{\boldsymbol{S}_{d,j}^{(k)}})) \qquad \text{(by Theorem 5.2)}$$

$$\leq O\left(\lambda w_G + n \log(n w_G)\right) +$$

$$\sum_{k=1}^{n} \sum_{j=1}^{2^{d+1}} \mathop{\mathbf{E}}_{\boldsymbol{S}_{d,j}} \mathrm{EM}_k(\mathrm{REC}(\boldsymbol{H}_{d+1,j}^{\boldsymbol{S}_{d,j}}), \mathrm{REC}(\boldsymbol{H}_{d+1,j}^{\boldsymbol{S}_{d,j}^{(k)}})). \qquad \text{(by } \phi^*_{\boldsymbol{H}_{d,j}} \leq w_{\boldsymbol{H}_{d,j}}/|V(\boldsymbol{H}_{d,j})|)$$

This implies that

$$\mathop{\mathbf{E}}_{\boldsymbol{H}_{d,j}} \sum_{k=1}^{n} \sum_{j=1}^{2^{d}} \mathrm{EM}_k(\mathrm{REC}(\boldsymbol{H}_{d,j}), \mathrm{REC}(\boldsymbol{H}_{d,j}^{(k)}))$$

$$\leq O\left(\lambda w_G + n \log(n w_G)\right) +$$

$$\sum_{k=1}^{n} \sum_{j=1}^{2^{d+1}} \mathop{\mathbf{E}}_{\boldsymbol{H}_{d+1,j}} \mathrm{EM}_k(\mathrm{REC}(\boldsymbol{H}_{d+1,j}), \mathrm{REC}(\boldsymbol{H}_{d+1,j}))$$

$$= O\left((D-d)(\lambda w_G + n \log(n w_G))\right)$$

as desired. $\qquad \square$

## D PROOFS OF SECTION 6

### D.1 THEORETICAL PROPERTIES OF SAMPLING

We first show that the distribution of $\mathrm{SAMPLING}(p, \boldsymbol{\pi})$ is equal to the one induced by $p$.

**Lemma D.1.** *For any $i \in \{1, \ldots, n\}$, we have $\Pr_{\boldsymbol{\pi}}[\mathrm{SAMPLING}(p, \boldsymbol{\pi}) = i] = p_i$.*

*Proof.* We note that if $t \in [q_{\sigma(j-1)}, q_{\sigma(j)}]$ for $j \leq i$, then we sample $\sigma(i)$ with probability $1/(n - j + 1)$. Then, the probability that we sample $\sigma(i)$ is

$$\sum_{j=1}^{i} \frac{q_{\sigma(j)} - q_{\sigma(j-1)}}{n - j + 1} = \sum_{j=1}^{i} \left( p_{\sigma(j)} - p_{\sigma(j-1)} \right) = p_{\sigma(i)}. \qquad \square$$

Next, we show that perturbing the vector $p \in \mathbb{R}^n_+$ does not change the output of SAMPLING with high probability over the random bits.

**Lemma D.2.** *For any $p, p' \in \mathbb{R}^n_+$ with $\sum_{i=1}^{n} p_i = \sum_{i=1}^{n} p'_i = 1$, we have*

$$\Pr_{\boldsymbol{\pi}}[\mathrm{SAMPLING}(p, \boldsymbol{\pi}) \neq \mathrm{SAMPLING}(p', \boldsymbol{\pi})] \leq \sum_{i=1}^{n} |p_i - p'_i|.$$

Note that the RHS is twice the total variation distance between distributions that output $i \in \{1, 2, \ldots, n\}$ with probability $p_i$ and $p'_i$. The proof is easy but technical.

*Proof of Lemma D.2.* For simplicity, we rename elements so that $p_1 \leq p_2 \leq \cdots \leq p_n$. It suffices to consider the case $p'_i = p_i + \delta$ and $p'_j = p_j - \delta$ for some $1 \leq i, j \leq n$ and $\delta > 0$ and the order of $p_i$'s are preserved, i.e., $p'_1 \leq p'_2 \leq \cdots \leq p'_n$, because the general case can be decomposed into a sequence of such cases.

We further assume that $i < j$ because the analysis for the other case is similar. Let $i_{\boldsymbol{\pi}} = \mathrm{SAMPLING}(p, \boldsymbol{\pi})$ and $i'_{\boldsymbol{\pi}} = \mathrm{SAMPLING}(p', \boldsymbol{\pi})$. We have $i_{\boldsymbol{\pi}} \neq i'_{\boldsymbol{\pi}}$ when we sample $t$ and $i$ such that $\min\{q_i, q'_i\} < t < \max\{q_i, q'_i\}$. The probability that such an event happens is $\frac{1}{n} \sum_{k=1}^{n} |q'_k - q_k|$, and our goal is to bound this sum.

We first note that $q'_k - q_k = 0$ for any $k \leq i - 1$. Also for any $k \in \{1, 2, \ldots, n\}$, we have

$$q_k = q_{k-1} + (n - k + 1)(p_k - p_{k-1}),$$
$$q'_k = q'_{k-1} + (n - k + 1)(p'_k - p'_{k-1}),$$

and hence we have

$$q'_k - q_k = q'_{k-1} - q_{k-1} + (n - k + 1)((p'_k - p'_{k-1}) - (p_k - p_{k-1})).$$

for any $k \in \{1, 2, \ldots, n\}$. In particular, this implies that

$$q'_i - q_i = (n - i + 1)((p'_i - p'_{i-1}) - (p_i - p_{i-1})) = (n - i + 1)\delta.$$

To analyze the sum, we consider the following three cases.

**Case $j = i + 1$:** In this case, we have

$$q'_j - q_j = q'_i - q_i + (n - j + 1)((p'_j - p'_i) - (p_j - p_i))$$
$$= (n - i + 1)\delta + (n - j + 1)(-2\delta) = -(n - j)\delta,$$
$$q'_{j+1} - q_{j+1}$$
$$= q'_j - q_j + (n - (j + 1) + 1)((p'_{j+1} - p'_j) - (p_{j+1} - p_j))$$
$$= -(n - j)\delta + (n - j)\delta = 0$$

and hence $q'_k - q_k = 0$ for any $k \geq j + 1$.

**Case $j = i + 2$:** In this case, we have

$$q'_{i+1} - q_{i+1}$$
$$= (q'_i - q_i) + (n - (i + 1) + 1)((p'_{i+1} - p'_i) - (p_{i+1} - p_i))$$
$$= (n - i + 1)\delta + (n - (i + 1) + 1)(-\delta) = \delta,$$
$$q'_j - q_j = q'_{i+1} - q_{i+1} + (n - j + 1)((p'_j - p'_{j-1}) - (p_j - p_{j-1}))$$
$$= \delta + (n - j + 1)(-\delta) = -(n - j)\delta,$$
$$q'_{j+1} - q_{j+1}$$
$$= q'_j - q_j + (n - (j + 1) + 1)((p'_{j+1} - p'_j) - (p_{j+1} - p_j))$$
$$= -(n - j)\delta + (n - (j + 1) + 1)\delta = 0$$

and hence $q'_k - q_k = 0$ for any $k \geq j + 1$.

**Case $j > i + 2$:** In this case, we have

$$q'_{i+1} - q_{i+1} = \delta,$$
$$q'_k - q_k = q'_{k-1} - q_{k-1} + (n - k + 1)((p'_k - p'_{k-1}) - (p_k - p_{k-1}))$$
$$= \delta \; (i + 2 \leq k \leq j - 1),$$
$$q'_j - q_j = q'_{j-1} - q_j + (n - j + 1)((p'_j - p'_{j-1}) - (p_j - p_{j-1}))$$
$$= \delta + (n - j + 1)(-\delta) = -(n - j)\delta,$$
$$q'_{j+1} - q_{j+1}$$
$$= q'_j - q_j + (n - (j + 1) + 1)((p'_{j+1} - p'_j) - (p_{j+1} - p_j))$$
$$= -(n - j)\delta + (n - (j + 1) + 1)\delta = 0$$

and hence $q'_k - q_k = 0$ for any $k \geq j + 1$.

In all the three cases, we have

$$\frac{1}{n} \sum_{k=1}^{n} |q'_k - q_k| = \frac{1}{n} \left( (n - i + 1)\delta + (n - j)\delta + (j - i - 1)\delta \right)$$

$$= \left( 2 - \frac{2i}{n} \right) \delta = \left( 1 - \frac{i}{n} \right) \sum_{k=1}^{n} |p'_k - p_k|. \qquad \square$$

## D.2 PROOF OF THEOREM 6.1

We consider a variant of SSC called SSC' (Algorithm 5) obtained from SSC by replacing the exponential mechanism with SAMPLING. The following holds:

---

**Algorithm 5:** Sparsest cut with a low average sensitivity under shared randomness

1  **Procedure** SSC'($G = (V, E, w), \lambda, \pi$)
2     **for** $\{i, j\} \in \binom{V}{2}$ *with* $i < j$ **do**
3        $S_{ij} \leftarrow \{k \in V : w(i, k) > w(j, k)\}$;
4        $\phi_G(i, j) \leftarrow \phi_G(S_{ij})$.
5     $Z \leftarrow 0$;
6     **for** $\{i, j\} \in \binom{V}{2}$ *with* $i < j$ **do**
7        $Z \leftarrow Z + \exp(-\lambda \phi_G(i, j))$.
8     **for** $\{i, j\} \in \binom{V}{2}$ *with* $i < j$ **do**
9        $p_{ij} \leftarrow \exp(-\lambda \phi_G(i, j))/Z$.
10    $(\boldsymbol{i}, \boldsymbol{j}) \leftarrow$ SAMPLING$(p, \pi)$;
11    **return** $S_{\boldsymbol{ij}}$.

---

**Lemma D.3.** *Let $G$ be a weighted graph with $n$ vertices, $\lambda > 0$. Then, the distribution of* SSC'$(G, \lambda, \boldsymbol{\pi})$ *over random bits $\boldsymbol{\pi}$ is equal to that of* SSC$(G, \lambda)$. *Moreover, we have*

$$\mathop{\mathbf{E}}_{\boldsymbol{\pi}} \left[ \frac{1}{n} \sum_{k \in V} 1[\text{SSC'}(G, \lambda, \boldsymbol{\pi}) \neq \text{SSC'}(G - k, \lambda, \boldsymbol{\pi})] \right]$$
$$= O\left( \frac{1}{n} \left( \lambda \phi_G^* + \log(n w_G) \right) \right),$$

*where $1[X]$ denotes the indicator of the event $X$.*

*Proof.* The first claim follows by Lemma D.1, and the second claim follows by combining Lemmas B.3 and D.2. $\qquad\square$

*Proof of Theorem 6.1.* Consider the algorithm SHC' obtained from SHC by replacing the calls of SSC in REC with SSC'. Combining the proof of Theorem 5.3 and Lemma D.3, we obtain the claim. $\qquad\square$

## E   ADDITIONAL EXPERIMENTAL RESULTS

### E.1   EVALUATION CRITERIA

In the experiments, we used Dendrogram Purity and Cohphenetic Correlation, in addition to Dasgupta score we reported in the main body of the paper.

**Dendrogram Purity**   Dendrogram Purity measures the quality of clustering using ground-truth class labels, which are unavailable in practice. For two points $x_i$ and $x_j$ belonging to the same class $k$, we define their *purity* $\texttt{pur}(i, j; k)$ by $\texttt{pur}(i, j; k) = |L_{ij} \cap C_k|/|L_{ij}|$, where $L_{ij}$ is the set of data points belonging to the subtree rooted at the lowest common ancestor of nodes that $x_i$ and $x_j$ belong to, and $C_k \subseteq X$ is the set of data points whose ground-truth labels are $k$. The Dendrogram Purity of $T$ is then defined as

$$\text{purity}(T) = \frac{2 \sum_{k=1}^{K} \sum_{i,j \in C_k} \texttt{pur}(i, j; k)}{\sum_{k=1}^{K} |C_k|(|C_k| - 1)}.$$

Dendrogram Purity takes a value close to one when the data points in the same class form a subtree in $T$, i.e., when they form a single cluster at a certain depth of the tree. On the other hand, it takes a value close to zero when data points in the same class are distributed to several distinct subtrees of $T$. Thus, a hierarchical clustering $T$ with larger $\text{impurity}(T)$ is considered ideal.

**Cohphenetic Correlation**   Let $m(i, j)$ be the depth of the root of the subtree rooted at the lowest common ancestor of nodes that $x_i$ and $x_j$ belong to. We also denote the distance between $x_i$ and

$x_j$ by $d(i,j) = \|x_i - x_j\|$. Cohphenetic Correlation is then defined as the negative correlation of $m(i,j)$ and $d(i,j)$,

$$\operatorname{corr}(T) = -\frac{\sum_{i<j}\left(m(i,j) - \bar{m}\right)\left(d(i,j) - \bar{d}\right)}{\sqrt{\sum_{i<j}\left(m(i,j) - \bar{m}\right)^2}\sqrt{\sum_{i<j}\left(d(i,j) - \bar{d}\right)^2}},$$

where $\bar{m} = \frac{1}{(n-1)(n-2)}\sum_{i<j}m(i,j)$ and $\bar{d} = \frac{1}{(n-1)(n-2)}\sum_{i<j}d(i,j)$. Cohphenetic Correlation takes a value close to one when distant data points are distributed to different subtrees so that their common ancestor to be shallow nodes. Thus, a hierarchical clustering $T$ with larger $\operatorname{corr}(T)$ is considered ideal.

### E.2 RESULTS

Figures 2, 4, and 5 show the results of the experiments when the data size is $n = 100$ and the tree depth is $D = 10$. Each figure shows the trade-offs between the average sensitivity and the average quality metric, Dasgupta Score, Dendrogram Purity, and Cohphenetic Correlation, respectively, with the similarity coefficient $\alpha$ varied to 1, 3, 10, and 30. The results of the baselines and SHC' for several different $\lambda$ are shown in different symbols and red lines, respectively. Because smaller Dasgupta Score is considered ideal, the lower left results are preferred as the clusterings with better quality and stability. For Dendrogram Purity and Cohphenetic Correlation, larger scores are considered ideal. Thus, the lower right results are preferred as the clusterings with better quality and stability.

We can find that the red lines of SHC' tend to lie in the lower left (or lower right) area of the figures. That is, SHC' with appropriately chosen $\lambda$ can attain a good trade-off with small average sensitivity and better quality metrics, as expected. By contrast, the agglomerative algorithms, except for `single`, tend to exhibit better quality metrics while incurring high average sensitivity. These methods are therefore good at producing high quality clusterings, while being sensitive to a small perturbation of the dataset. The result of `single` is exceptional, exhibiting worse quality metrics with small average sensitivity. We observed that `single` tends to produce highly unbalanced clusterings because `single` split the dataset into small and large clusters. Although such a split is less sensitive to the perturbation of the dataset and has smaller average sensitivity, the quality of clustering is poor as shown in the figures. SHC' provides a way to balance the quality of the clustering and its average sensitivity by tuning $\lambda$ upon the user demand.

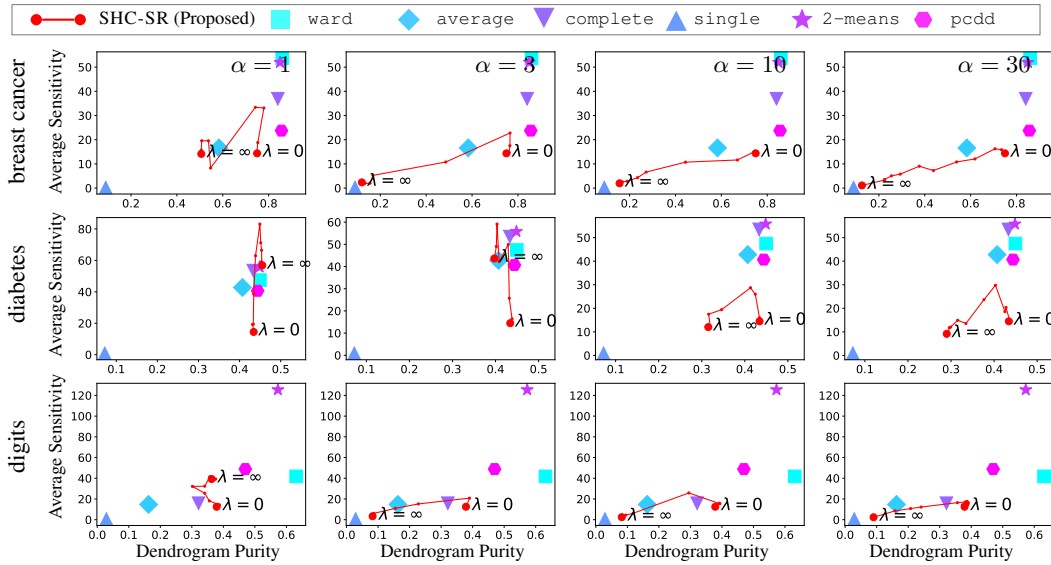

Figure 4: Trade-offs between average sensitivity and Dendrogram Purity for the data size $n = 100$, depth $D = 10$, and $\alpha = 1, 3, 10, 30$. The results of SHC-SR are shown in red lines displaying the results for several different $\lambda \in \{0, 10^{-3}, \ldots, 10^6, \infty\}$.

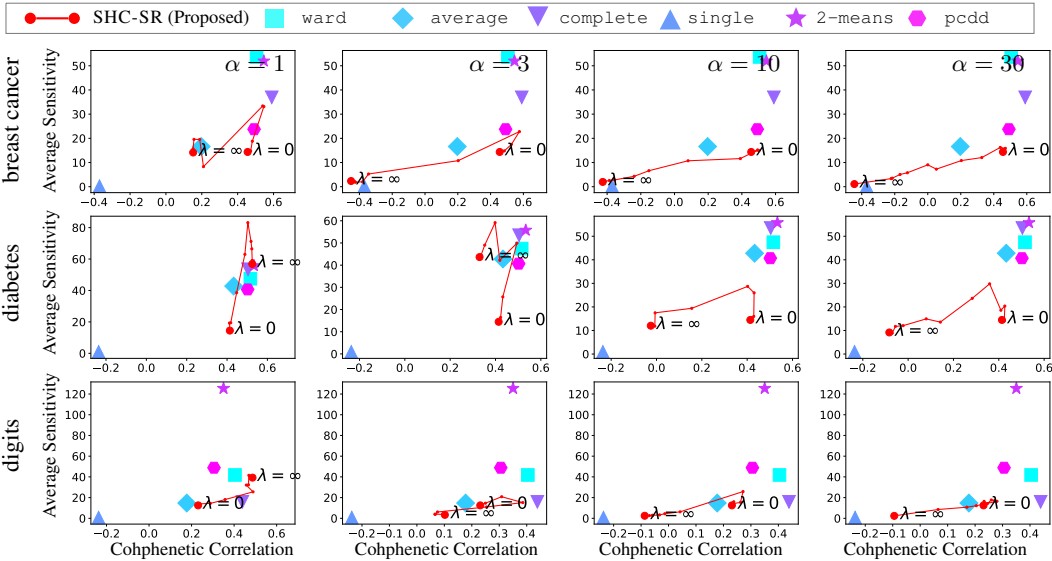

Figure 5: Trade-offs between average sensitivity and Cohphenetic Correlation for the data size $n = 100$, depth $D = 10$, and $\alpha = 1, 3, 10, 30$. The results of SHC-SR are shown in red lines displaying the results for several different $\lambda \in \{0, 10^{-3}, \ldots, 10^6, \infty\}$.

### E.3    RESULTS WITH DIFFERENT DATA SIZE

Figures 6, 7, and 8 show the results of the experiments when the data size is $n = 300$ and the tree depth is $D = 10$. Similarly, Figures 9, 10, and 11 show the results of the experiments when the data size is $n = 300$ and the tree depth is $D = 10$.[5] In these figures, similar to $n = 100$ (Figures 2, 4, 5), the red lines of SHC' tend to lie in the lower left (or lower right) area of the figures, indicating that SHC' with appropriately chosen $\lambda$ can attain a good trade-off even for different data sizes.

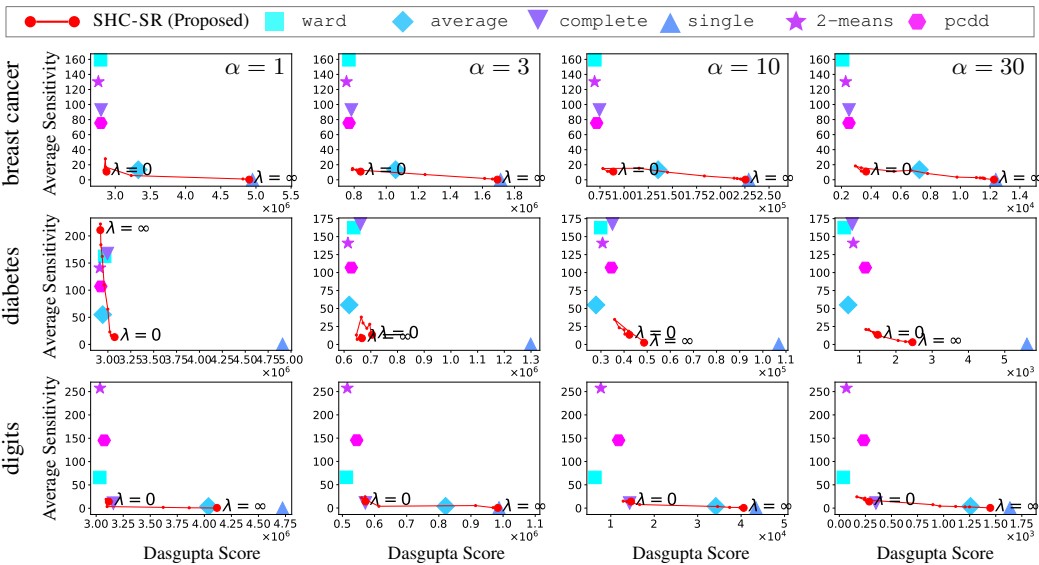

Figure 6: Trade-offs between average sensitivity and Dasgupta Score for the data size $n = 300$, depth $D = 10$, and $\alpha = 1, 3, 10, 30$. The results of SHC-SR are shown in red lines displaying the results for several different $\lambda \in \{0, 10^{-3}, \ldots, 10^6, \infty\}$.

---

[5]The results on diabetes is omitted because its original data size is smaller than 500.

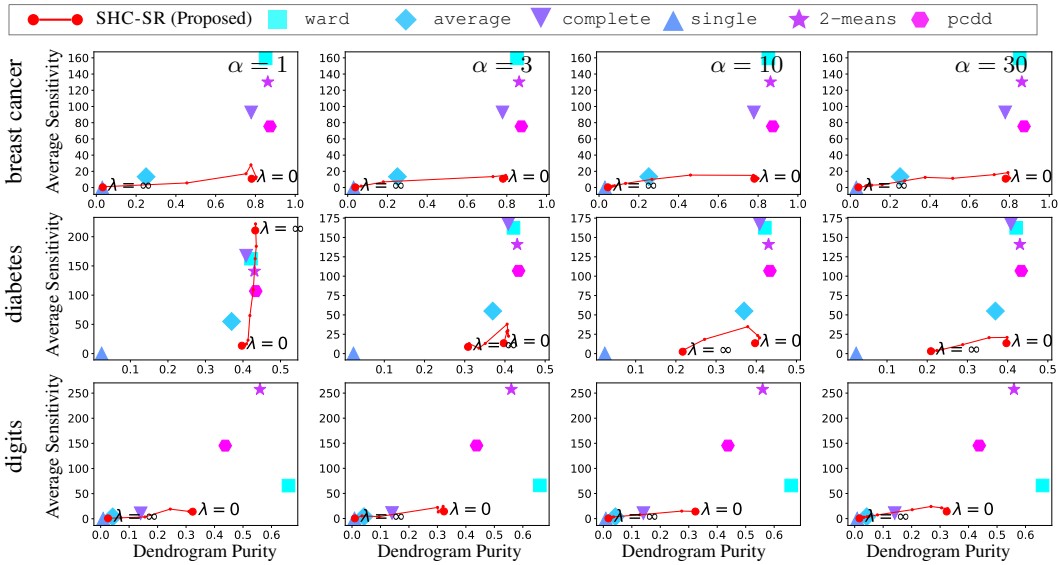

Figure 7: Trade-offs between average sensitivity and Dendrogram Purity for the data size $n = 300$, depth $D = 10$, and $\alpha = 1, 3, 10, 30$. The results of SHC-SR are shown in red lines displaying the results for several different $\lambda \in \{0, 10^{-3}, \dots, 10^6, \infty\}$.

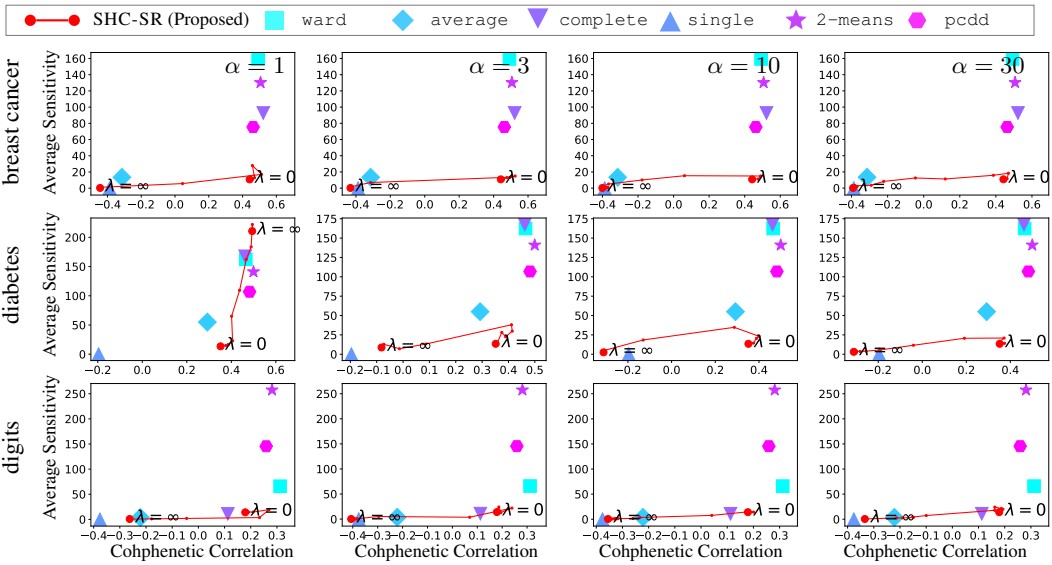

Figure 8: Trade-offs between average sensitivity and Cohphenetic Correlation for the data size $n = 300$, depth $D = 10$, and $\alpha = 1, 3, 10, 30$. The results of SHC-SR are shown in red lines displaying the results for several different $\lambda \in \{0, 10^{-3}, \dots, 10^6, \infty\}$.

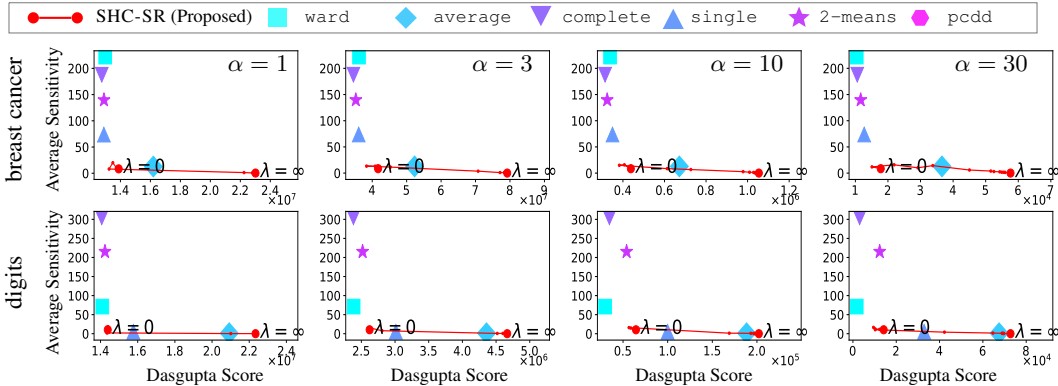

Figure 9: Trade-offs between average sensitivity and Dasgupta Score for the data size $n = 500$, depth $D = 10$, and $\alpha = 1, 3, 10, 30$. The results of SHC-SR are shown in red lines displaying the results for several different $\lambda \in \{0, 10^{-3}, \ldots, 10^6, \infty\}$.

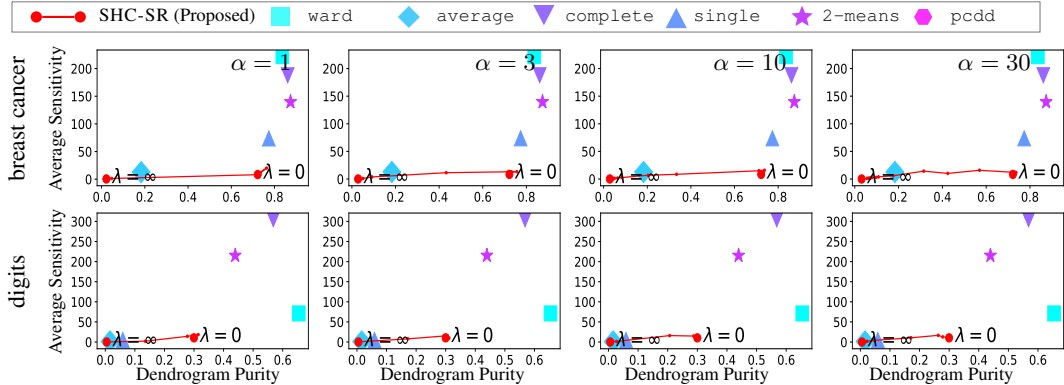

Figure 10: Trade-offs between average sensitivity and Dendrogram Purity for the data size $n = 500$, depth $D = 10$, and $\alpha = 1, 3, 10, 30$. The results of SHC-SR are shown in red lines displaying the results for several different $\lambda \in \{0, 10^{-3}, \ldots, 10^6, \infty\}$.

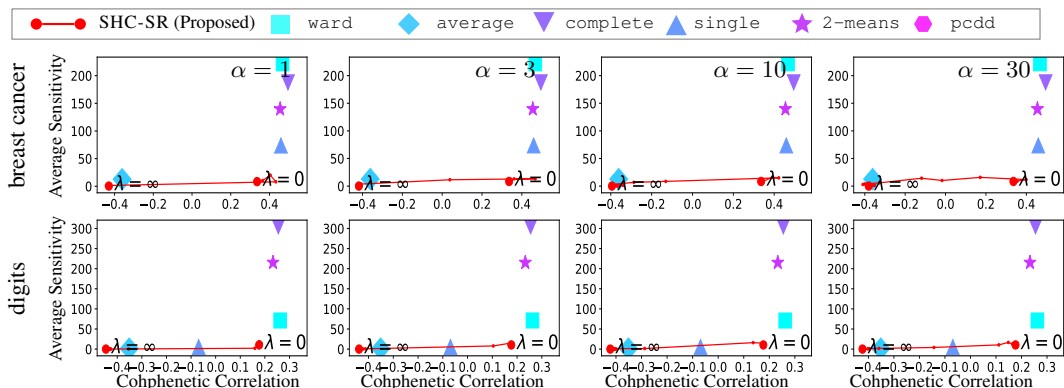

Figure 11: Trade-offs between average sensitivity and Cophenetic Correlation for the data size $n = 500$, depth $D = 10$, and $\alpha = 1, 3, 10, 30$. The results of SHC-SR are shown in red lines displaying the results for several different $\lambda \in \{0, 10^{-3}, \ldots, 10^6, \infty\}$.

### E.4 RESULTS WITH DIFFERENT TREE DEPTH

Figures 12, 13, and 14 show the results of the experiments when the data size is $n = 100$ and the tree depth is $D = 20$. In these figures, similar to $D = 10$ (Figures 2, 4, and 5), the red lines of SHC' tend to lie in the lower left (or lower right) area of the figures, indicating that SHC' with appropriately chosen $\lambda$ can attain a good trade-off even for different data sizes.

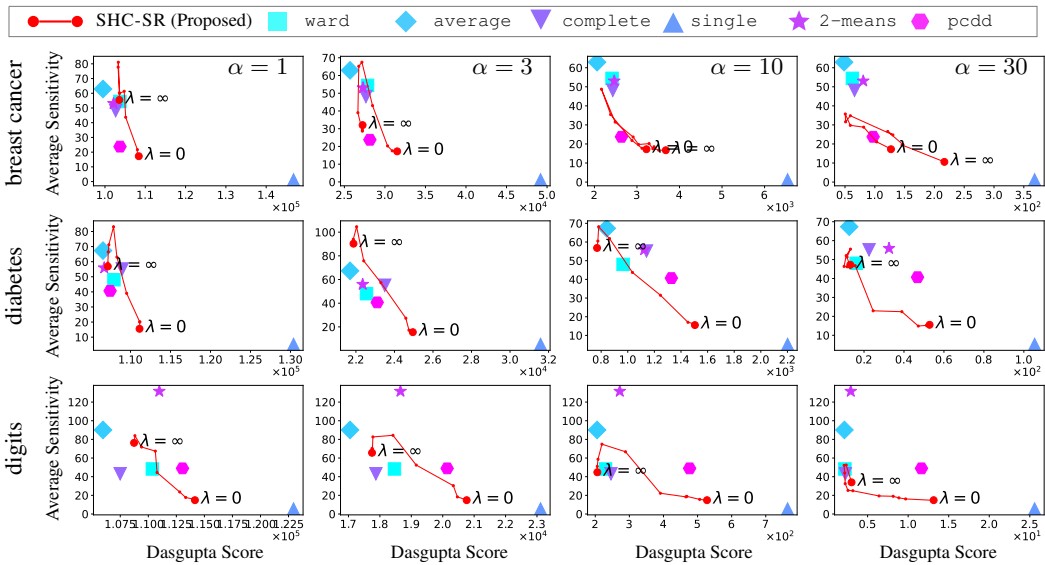

Figure 12: Trade-offs between average sensitivity and Dasgupta Score for the data size $n = 100$, depth $D = 20$, and $\alpha = 1, 3, 10, 30$. The results of SHC-SR are shown in red lines displaying the results for several different $\lambda \in \{0, 10^{-3}, \dots, 10^6, \infty\}$.

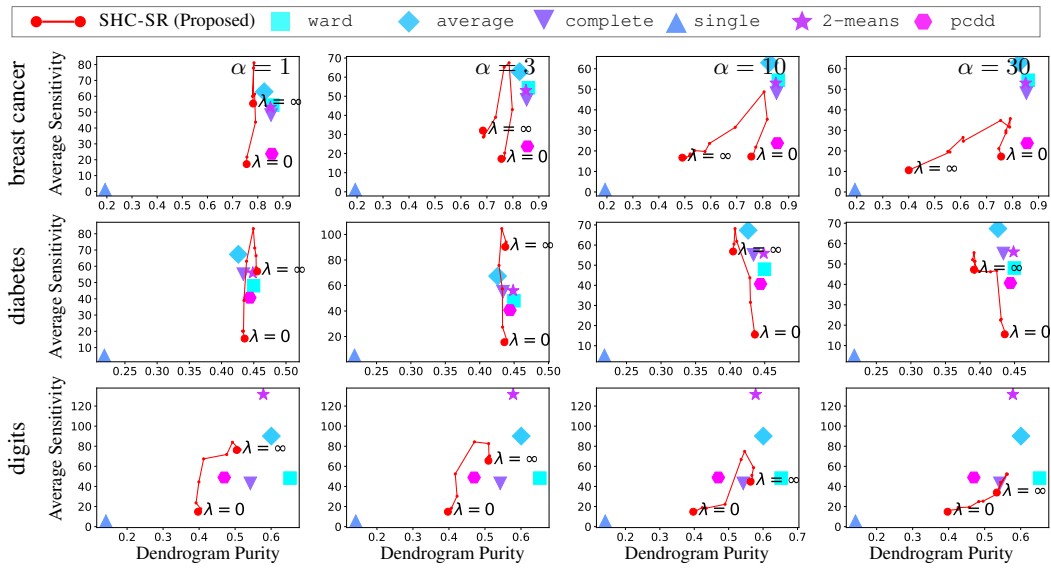

Figure 13: Trade-offs between average sensitivity and Dendrogram Purity for the data size $n = 100$, depth $D = 20$, and $\alpha = 1, 3, 10, 30$. The results of SHC-SR are shown in red lines displaying the results for several different $\lambda \in \{0, 10^{-3}, \ldots, 10^6, \infty\}$.

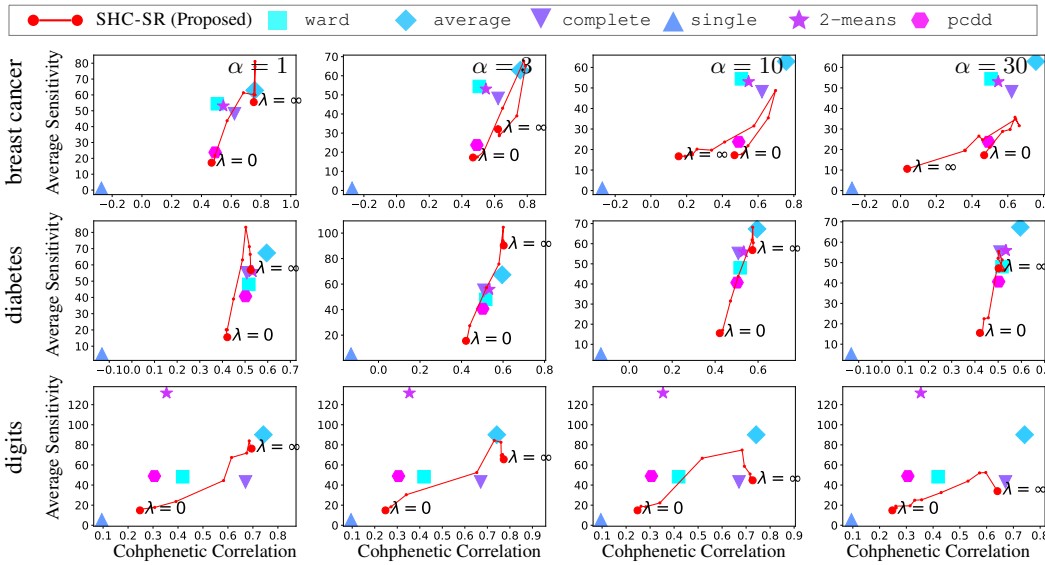

Figure 14: Trade-offs between average sensitivity and Cohphenetic Correlation for the data size $n = 100$, depth $D = 20$, and $\alpha = 1, 3, 10, 30$. The results of SHC-SR are shown in red lines displaying the results for several different $\lambda \in \{0, 10^{-3}, \ldots, 10^6, \infty\}$.

## E.5 Results with Errors

Figures 15, 16, and 17 show the results with errors when the data size is $n = 100$ and the tree depth is $D = 10$. The ellipsoids denote standard deviations over 10 trials. For SHC-SR, the ellipsoids are shown for $\lambda = 0$ and $\lambda = \infty$ for the visibility purpose.

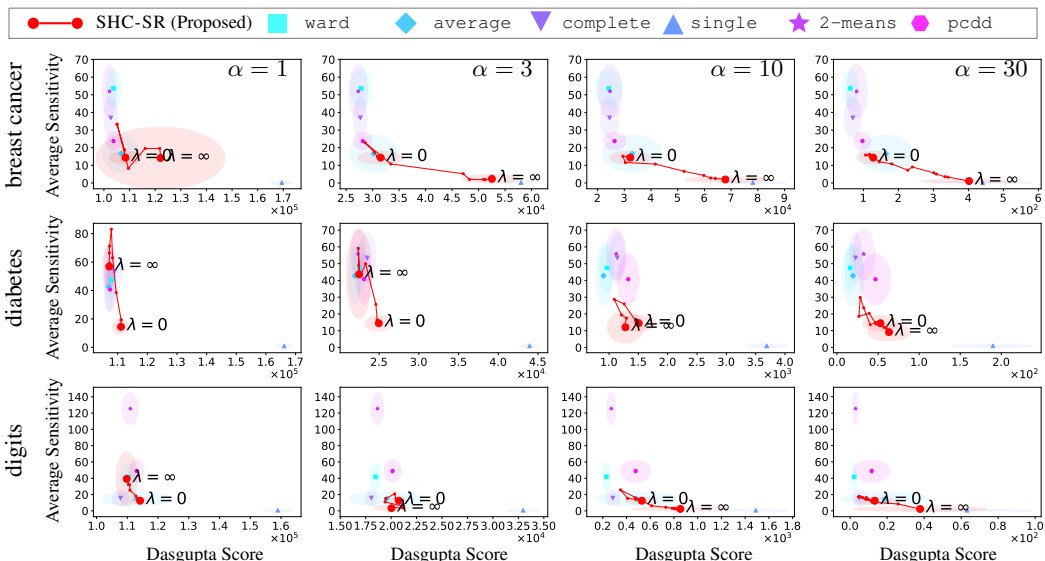

Figure 15: Trade-offs between average sensitivity and Dasgupta Score for the data size $n = 100$, depth $D = 10$, and $\alpha = 1, 3, 10, 30$. The results of SHC-SR are shown in red lines displaying the results for several different $\lambda \in \{0, 10^{-3}, \dots, 10^6, \infty\}$.

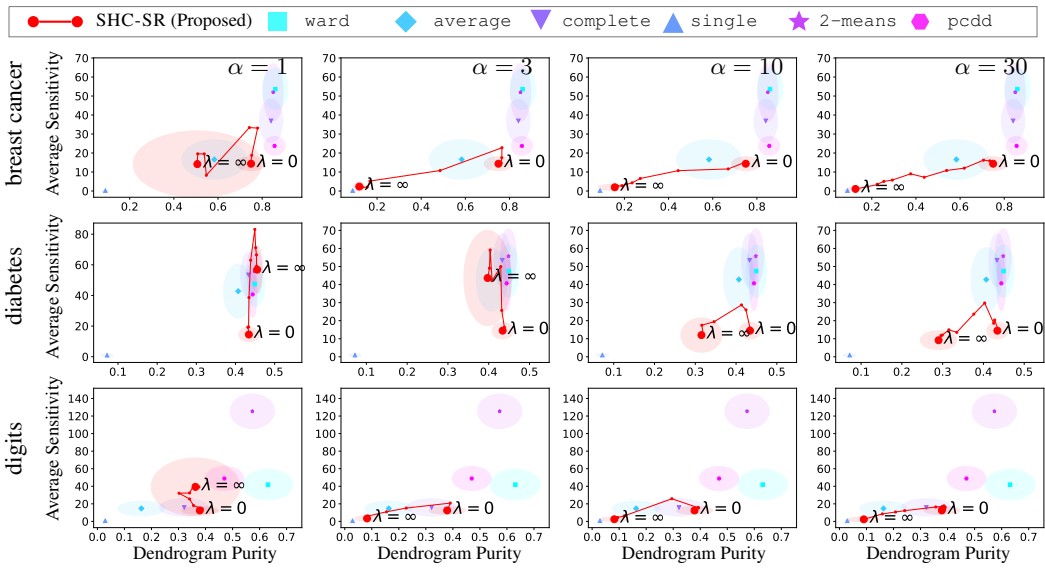

Figure 16: Trade-offs between average sensitivity and Dendrogram Purity for the data size $n = 100$, depth $D = 10$, and $\alpha = 1, 3, 10, 30$. The results of SHC-SR are shown in red lines displaying the results for several different $\lambda \in \{0, 10^{-3}, \ldots, 10^6, \infty\}$.

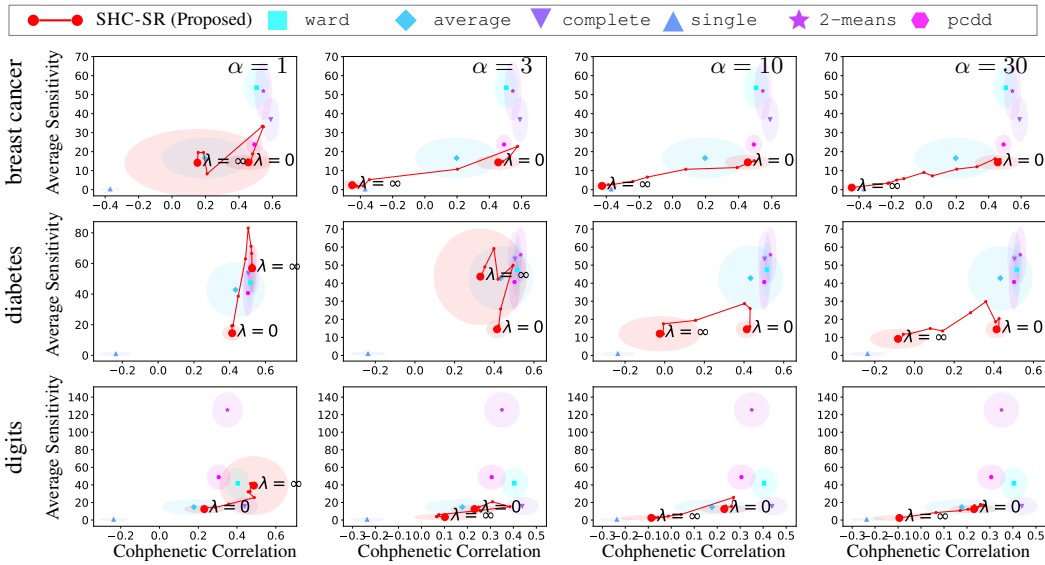

Figure 17: Trade-offs between average sensitivity and Cohphenetic Correlation for the data size $n = 100$, depth $D = 10$, and $\alpha = 1, 3, 10, 30$. The results of SHC-SR are shown in red lines displaying the results for several different $\lambda \in \{0, 10^{-3}, \ldots, 10^6, \infty\}$.

