# OpenReview forum: "Average Sensitivity of Hierarchical Clustering"
_ICLR.cc/2024/Conference — Submitted to ICLR 2024_

### Official Review · Reviewer_essW · 2023-10-22

**Soundness:** 3 good
**Presentation:** 3 good
**Contribution:** 2 fair
**Rating:** 6
**Confidence:** 4

**Summary:**

The average sensitivity of a randomized algorithm measures the distance between the distributions of the output of the algorithm when a random element of the input is deleted, where the distance is often the total variation distance or the earth mover distance. It can often be advantageous to minimize the average sensitivity of an algorithm so that small changes to the input may only result in small changes to the output.

This paper studies the average sensitivity of algorithms for hierarchical clustering, defining a distance between hierarchical clusterings of a graph by the sizes of the symmetric differences at each level. The paper first proposes an algorithm for stable hierarchical clustering, which is based off recursively performing a stable sparsest cut subroutine that crucially utilizes the exponential mechanism from differential privacy. The paper then shows that if the randomness between multiple instances of the algorithm is shared, then a smaller average sensitivity can be induced by using a permutation-based sampling algorithm to replace the exponential mechanism. Finally, the paper performs a number of experiments showing tradeoffs between average sensitivity and the performance of the hierarchical clustering algorithms under the Dasgupta objective.

**Strengths:**

- Although average sensitivity is less studied than worst-case sensitivity for privacy, I believe the problem can still be well-motivated from the perspective of average-case data perturbations and therefore relevant to a theoretical subcommunity of ICLR
- The paper provides rigorous analysis upper bounding the average sensitivity of the proposed algorithms
- The experiments are performed over multiple datasets and clearly show a trade-off between average sensitivity and clustering cost
- The problem setup, the algorithm, and the intuition are all well-written and accessible to a general audience

**Weaknesses:**

- Applications of methods from differential privacy, i.e., exponential mechanism, may not be as surprising as initially seemed, due to connections to the notion of worst-case sensitivity in differential privacy
- The main approximation guarantee for the stable-on-average hierarchical clustering algorithm seems to be in terms of the expected size of the sparsest cut, rather than the expected cost of the clustering
- Other hierarchical clustering objectives such as the Moseley-Wang or Cohen-Addad objectives are not considered

**Questions:**

- What can be said about the approximation guarantees of the main algorithm in terms of the Dasgupta objective? Can anything be said about the approximation guarantees of the algorithm given an $\alpha$-approximation algorithm for sparsest cut?
- Is it true that the random cut algorithm that provides a good approximation to the Moseley-Wang objective would also have low average sensitivity?

---

> ### Author Response · Authors · 2023-11-22
> **Reply**
>
> We thank the reviewer for helpful comments. We provide detailed answers to comments and questions below.
>
> > Applications of methods from differential privacy, i.e., exponential mechanism, may not be as surprising as initially seemed, due to connections to the notion of worst-case sensitivity in differential privacy
>
> Differential privacy (DP) corresponds to measuring the sensitivity using TV distance with $\max\_{x \in X}$ instead of $\frac{1}{n} \sum\_{x \in X}$ in Eq. (1), i.e., DP is more pesimistic than Average Sensitivity (AS). Because of this relation, one can derive a trivial bound like $\mathrm{AS} \le \mathrm{DP} \cdot n$. In general, ensuring the stability with respect to DP inccurs worse complexity.
>
> > The main approximation guarantee for the stable-on-average hierarchical clustering algorithm seems to be in terms of the expected size of the sparsest cut, rather than the expected cost of the clustering
>
> This is correct. Here, we would like to remind that the sparsest cut is related to Dasgupta score.
>
> > Other hierarchical clustering objectives such as the Moseley-Wang or Cohen-Addad objectives are not considered
>
> Moseley-Wang is essentially the same as Dasguta score, while Cohen-Addad is a generalization of Dasguta score. The question when using Cohen-Addad is on the choice of the function $g$, where Dasguta score corresponds to a specific choice of $g$. We there think the use of Moseley-Wang or Cohen-Addad do not add much on our results.
>
> > What can be said about the approximation guarantees of the main algorithm in terms of the Dasgupta objective? Can anything be said about the approximation guarantees of the algorithm given an $\alpha$-approximation algorithm for sparsest cut?
>
> Theorem 12 of (Dasgupta, 2016) showed the $O(\alpha \log n)$ approximation.
>
> > Is it true that the random cut algorithm that provides a good approximation to the Moseley-Wang objective would also have low average sensitivity?
>
> The random cut will induce zero average sensitivity. Thus, the overall average sensitivity will also be zero. However, the 1/3-approximation of Moseley-Wang may not be that useful in practice because random cut will not provide any interesting clusters.

---

> ### Comment · Reviewer_essW · 2023-11-22
>
> Thanks for the response.
>
> I'm not sure I understand the notion that Moseley-Wang is essentially the same as the Dasgupta score, while Cohen-Addad is a generalization of the Dasgupta score, especially if we provide approximation guarantees in terms of the clustering cost, rather than the size of the cut. For example, you noted that Mosely-Wang has 1/3 approximation algorithm that may not be useful in practice, while achieving constant-factor approximation for the Dasgupta objective seems challenging.
>
> Would you be able to elaborate?

---

> > ### Author Response · Authors · 2023-11-23
> > **Re: Official Comment by Reviewer essW**
> >
> > > Moseley-Wang is essentially the same as the Dasgupta score
> >
> > Moseley-Wang is the "dual" of the Dasgupta score.
> > Let $\mathrm{cost}(T)$ be the Dasgupta score and $\mathrm{rev}(T)$ be the score proposed by Moseley-Wang.
> > Then, Moseley-Wang has shown that $\mathrm{cost}(T) + \mathrm{rev}(T) = n \sum\_{i, j} w(i, j)$ .
> > Thus, better (smaller) $\mathrm{cost}(T)$ indicates better (larger) $\mathrm{rev}(T)$.
> >
> > > Cohen-Addad is a generalization of the Dasgupta score
> >
> > Cohen-Addad showed that one can generalize the Dasgupta score as
> > $$
> > \Gamma(T) = \sum_{\mathrm{node} \in T} \left(\sum\_{x \in \mathrm{left}(\mathrm{node})} \sum\_{y \in \mathrm{right}(\mathrm{node})} w(x, y) \right) g(|\mathrm{left}(\mathrm{node})|, |\mathrm{right}(\mathrm{node})|)
> > $$
> > where setting $g(a, b) = a + b$ yields the the Dasgupta score.
> >
> > > provide approximation guarantees in terms of the clustering cost, rather than the size of the cut
> >
> > As you noted, we will be able to obtain some approximation guarantees for useless algorithms (such as random cut). The challenge is to provide both the approixamtion and stability guarantess for useful algorithms that can produce high-quality clustering, which is unfurtunately beyond the scope of the current paper.

---

> > > ### Comment · Reviewer_essW · 2023-12-04
> > >
> > > Thank you for the response -- I acknowledge having read the follow-up.

---

### Official Review · Reviewer_8UiG · 2023-10-27

**Soundness:** 2 fair
**Presentation:** 2 fair
**Contribution:** 2 fair
**Rating:** 5
**Confidence:** 4

**Summary:**

In this paper, the authors study the average sensitivity of hierarchical clustering. In the paper, two hierarchical clustering algorithms that consider the average sensitivity are proposed, noted as SHC and SHC-SR. The proposed algorithms are theoretically analyzed. The SHC is proved to have low average sensitivity, and the SHC-SR is proved to have average sensitivity under shared randomness.

**Strengths:**

1.	The paper is well-written, organized logically, and has a strong theoretical foundation.
2.	This paper exhibits a degree of innovation in considering sensitivity within hierarchical clustering.

**Weaknesses:**

1.	The significance of this paper needs further clarification, especially regarding the unclear relationship between sensitivity and clustering performance. As illustrated in the example in Figure 1, the 10th sample can be considered as a boundary or noise sample, while the 4th sample can be considered as a central sample. These samples have varying impacts on the clustering tree. It can be anticipated that removing the 10th sample will result in minimal changes to the hierarchical tree, whereas removing the 4th sample will lead to significant changes. Why stable results are needed?
2.	The experimental analysis appears to be insufficient. Firstly, the experiments are conducted on three benchmark datasets and a single set of real-world data, which results in a relatively small dataset size. Secondly, common external clustering performance evaluation metrics, such as NMI, ARI, and ACC, are not employed in the experiments. Lastly, there is no clear discernible consistent trend evident from Figure 2 and the related figures in the supplementary materials.

**Questions:**

1.	Why is sensitivity crucial for hierarchical clustering?

---

> ### Author Response · Authors · 2023-11-22
> **Reply**
>
> We thank the reviewer for helpful comments. We provide detailed answers to comments and questions below.
>
> > Why stable results are needed?
>
> Hierarchical clustering is used to understand the data visually. If a cluster structure changes drastically, the users will be confused whether their data understanding is truly reliable or only a noisy artifact induced by the unstable clustering algorithm. The stability is essential for reliable data understading.
>
> > The experimental analysis appears to be insufficient. Firstly, the experiments are conducted on three benchmark datasets and a single set of real-world data, which results in a relatively small dataset size.
>
> The purpose of hierarchical clustering is to visualize the relationships between the data points. Visualization of thousands of data points are generally hard for users to inspect. Because of this reason, hierarchical clustering is better suited for the cases where the number of data points are not too large.
>
> > Secondly, common external clustering performance evaluation metrics, such as NMI, ARI, and ACC, are not employed in the experiments.
>
> We used the metrics tailored specifically for hierarchical clustering, which are Dasgupta score, Dendrogram purity, and Cophenetic correlation. We believe these are appropriate choices for assessing the proposed method, rather than using generic metircs.
>
> We nevertheless agree with the importance of evaluating the proposed method using several different metrics. However, we have to say that it is not possible to evaluate the method with (possibly) tens or hundreds of metrics. We therefore believe that the exhaustive evaluation is not the duty. Our responsibility is to relase the code so that any users with interest in alternative metrics can perform evaluations by modifying the provided code.
>
> > Lastly, there is no clear discernible consistent trend evident from Figure 2 and the related figures in the supplementary materials.
>
> We do not say that there is a discernible consistent trend in the results. Our bound $O(\lambda Dn)$ is the worst-case upper bound. It is sometimes the case that the worst-case bound and practical performances have some gaps. Our results indicate that using a slightly larger $\lambda$ can result in better stability in some cases.

---

### Official Review · Reviewer_qQY7 · 2023-10-28

**Soundness:** 2 fair
**Presentation:** 2 fair
**Contribution:** 2 fair
**Rating:** 3
**Confidence:** 4

**Summary:**

The paper proposes a method for hierarchical clustering that is robust to deletion of data points. Its stability is supported by theoretical results.

**Strengths:**

Hierarchical clustering is known to be unstable w.r.t. to removal of data points. The paper formalizes the concept of average sensitivity for hierarchical clustering methods that is used to assess their stability to such removal.

Algorithm SHC for stable hierarchical clustering is introduced whose average sensitivity is $O(\lambda D n)$ with $\lambda$ being an input parameter for exponential mechanism, $D$ depth of hierarchical clustering and $n$ the number of data samples.

**Weaknesses:**

SHC has time complexity $O(D n^3)$. This makes it impractical to moderate to large size data sets, which is perhaps the reason why experiments are conducted on mostly small data sets. I would like to see a more efficient algorithm, or at least an approximate one with proven theoretical bound.

The average sensitivity of SHC is $O(\lambda D n)$, which can be made small by setting $\lambda \ll 1$. In the experiments, this is not always the case, for instance in Section 8. This signifies that SHC in practice doesn't really have a low average sensitivity.

The data sets chosen in the experiments are way too small. This makes the results not so convincing.

**Questions:**

SHC's time complexity is quadratic in data size. Could this be alleviated?

Why was $\lambda > 1$ used in the experiments?

---

> ### Author Response · Authors · 2023-11-22
> **Reply**
>
> We thank the reviewer for helpful comments. We provide detailed answers to comments and questions below.
>
> > SHC has time complexity $O(Dn^3)$. This makes it impractical to moderate to large size data sets, which is perhaps the reason why experiments are conducted on mostly small data sets. I would like to see a more efficient algorithm, or at least an approximate one with proven theoretical bound.
>
> We would like to remind that the typical agglomerative methods also run in $O(n^3)$ time. Because small $D$ is typicall chosen, the $O(Dn^3)$ time of the propsoed method is comparable with the existing methods. We also observed in the experiments that the proposed methos can run faster than the agglomerative methods for $D \le 10$.
>
> > The average sensitivity of SHC is $O(\lambda Dn)$, which can be made small by setting $\lambda \ll 1$. In the experiments, this is not always the case, for instance in Section 8. This signifies that SHC in practice doesn't really have a low average sensitivity.
>
> Our bound $O(\lambda Dn)$ is the worst-case upper bound. It is sometimes the case that the worst-case bound and practical performances have some gaps. Our results indicate that using a slightly larger $\lambda$ can result in better stability in some cases.
>
> > The data sets chosen in the experiments are way too small. This makes the results not so convincing.
>
> The purpose of hierarchical clustering is to visualize the relationships between the data points. Visualization of thousands of data points are generally hard for users to inspect. Because of this reason, hierarchical clustering is better suited for the cases where the number of data points are not too large.
>
> > SHC's time complexity is quadratic in data size. Could this be alleviated?
>
> The answer is no because SSC requires $O(k^3)$ time for the size $k$ input.  Here,  we would like to remind that this time complexity is comparable with the existing agglomerative methods.
>
> > Why was $\lambda > 1$ used in the experiments?
>
> $\lambda \to 0$ corresponds to random sparse cut while $\lambda \to \infty$ corresponds to greedy sparse cut. In the experiments, we are interested in the middle of these two extremes.

---

### Official Review · Reviewer_seUc · 2023-10-31

**Soundness:** 3 good
**Presentation:** 3 good
**Contribution:** 1 poor
**Rating:** 3
**Confidence:** 3

**Summary:**

This paper considers the design of algorithms for hierarchical clustering whose output does not change significantly if a random data point is deleted from the input.  Such an algorithm is said to have low average sensitivity.  After formally defining average sensitivity for hierarchical clustering algorithms the paper designs and analyzes such an algorithm with low sensitivity which uses the exponential mechanism from differential privacy in a recursive splitting procedure.  In particular, the bound on the average sensitivity for the algorithm is $O(D \lambda w_G / n + D \log(n w_G))$, where $D$ is an input parameter controlling the maximum depth, $\lambda$ is an input parameter for the exponential mechanism, and $w_G$ is the total weight in a weighted graph constructed by the algorithm (each such weight is in $[0,1]$, hence we get the bound $w_G = O(n^2)$).  This is extended to the case of shared randomness, i.e., the case in which we fix the string of random bits for both the original input and the input with a random data point deleted.  Finally, the paper complements the theoretical results with a thorough experimental analysis of the proposed algorithm.

**Strengths:**

- For the most part, the paper is cleanly written and easy to read.
- The algorithms and analysis are well motivated and clearly described.
- Experimental results are promising

**Weaknesses:**

My main issue with this paper as it is currently written is in the formulation.  The definition of average sensitivity seems to only require that the output distribution for similar inputs are in some sense close.  While the proposed algorithms are certainly non-trivial its not clear to me that:
1. a similar guarantee cannot be given for a simpler algorithm, and
2. the proposed algorithms are doing something desirable in the context of hierarchical clustering

For the first point, it would help to see some notion of lower bound to better understand the complexity of the problem.  Expounding on the second point further, I think it would help if the algorithm was also constrained to perform well on some notion of utility, e.g., the Dasgupta cost function that is used as a metric in the experiments.  I think it would greatly strengthen the paper to consider the tradeoff between average sensitivity and some notion of clustering cost or utility (again, the Dasgupta cost may be a good candidate to consider).

**Questions:**

- The proposed algorithm makes use of the exponential mechanism from differential privacy.  How does average sensitivity relate to differential privacy?  Does one notion imply the other or vice versa?  Overall, these two notions both seem to capture the intuitive idea of "the output distribution for similar inputs should be similar".

- The average sensitivity seems to scale with the maximum allowable depth $D$.  Why not just set $D = 1$, or some other small value?  It feels like there should be a tension with some notion of clustering quality that benefits from using a tree with more depth.

- Can the quality of the proposed algorithm be analyzed in terms of some cost or utility function (e.g., Dasgupta's cost function or the utility function due to Moseley and Wang)?

---

> ### Author Response · Authors · 2023-11-22
> **Reply**
>
> We thank the reviewer for helpful comments. We provide detailed answers to comments and questions below.
>
> > While the proposed algorithms are certainly non-trivial its not clear to me that:
> > 1. a similar guarantee cannot be given for a simpler algorithm, and
> > 2. the proposed algorithms are doing something desirable in the context of hierarchical clustering
> > it would help to see some notion of lower bound to better understand the complexity of the problem
> > it would help if the algorithm was also constrained to perform well on some notion of utility, e.g., the Dasgupta cost function that is used as a metric in the experiments
>
> Unfortunately, we do not have a rigid answer to this point. The complexity analysis requires an approximation guarantee for the sparsest cut problem. However, the known approximation algorithm are computationally quite demanding even though they are polynomial time, which can result in $O(D \cdot \mathrm{poly}(n))$ time.
>
> > The proposed algorithm makes use of the exponential mechanism from differential privacy. How does average sensitivity relate to differential privacy? Does one notion imply the other or vice versa? Overall, these two notions both seem to capture the intuitive idea of "the output distribution for similar inputs should be similar".
>
> Differential privacy (DP) corresponds to measuring the sensitivity using TV distance with $\max\_{x \in X}$ instead of $\frac{1}{n} \sum\_{x \in X}$ in Eq. (1), i.e., DP is more pesimistic than Average Sensitivity (AS). Because of this relation, one can derive a trivial bound like $\mathrm{AS} \le \mathrm{DP} \cdot n$. In general, ensuring the stability with respect to DP inccurs worse complexity.
>
> > The average sensitivity seems to scale with the maximum allowable depth $D$. Why not just set $D=1$, or some other small value? It feels like there should be a tension with some notion of clustering quality that benefits from using a tree with more depth.
>
> $D$ is the depth of the hierarchical clustering, which is determined based on the users’ preference. $D=1$ induces only one splitting of the daset to two clusters, which is generally not sufficient for the clustering purpose.
>
> > Can the quality of the proposed algorithm be analyzed in terms of some cost or utility function (e.g., Dasgupta's cost function or the utility function due to Moseley and Wang)?
>
> For the analysis, we need an approximation guarantee for the sparsest cut problem. The typical approximation algorithms are LP (Leighton-Rao) and SDP (Arora-Rao-Vazirani). However, the rounding operation in LP will not be stable, and SDP is generally too slow. In the current state, we do not have a good way to ensure the approximation and stability guarantees in the same time.

---

> > ### Comment · Reviewer_seUc · 2023-11-22
> >
> > Thanks for the response.  I understand better, but I am not quite satisfied by the responses regarding the depth parameter $D$ and the lack of approximation/quality analysis.

---

### Meta-Review · Area_Chair_UFXW · 2023-12-04

**Metareview:**

There were several weaknesses pointed out by the reviewers.  In particular, the novelty of the paper isn't clear and how the theoretical results fit in with the current work on hierarchical clustering was no fully developed.   Overall, the paper may be looking into an interesting direction, but the ideas need to be further developed.

**Justification For Why Not Higher Score:**

There was not sufficient interest in the results and the novelty is unlcear.  Further, there were several questions on how this work fits into existing theoretical work that need further explanation.

**Justification For Why Not Lower Score:**

N/A

---

### Decision · Program_Chairs · 2024-01-16

Reject